# When Can We Trust Survival Model Evaluation?

**Ghanem Bahrini** [1 2]   **Sébastien Razakarivony** [1]   **Jean-François Dupuy** [2]   **Valérie Garès** [3]   **Morgane Barbet-Massin** [1]

## Abstract

Evaluating survival models under censoring is inherently challenging, yet standard evaluation practices are often applied without explicitly assessing how censoring distorts metric reliability. Performing a large experimental study, we analyze and quantify how survival evaluation metrics are affected in fundamentally different ways by the censoring rate and the censoring mechanism. Using a controlled semi-synthetic framework, we vary both the censoring mechanism (administrative, independent, covariate-dependent) and the censoring rate, and compare standard evaluations based on censored data with oracle evaluations using fully observed event times. This controlled setting enables us to quantify distortions along two complementary axes: numerical bias and preservation of model ranking. Across datasets and metric families, we find that censoring induces systematic, mechanism-dependent distortions. Moderate numerical bias, if not properly addressed, can lead to unreliable model comparison as censoring increases. These findings reveal fundamental limitations of common benchmarking practices and call for more careful interpretation of survival evaluation under realistic censoring.

## 1. Introduction

Time-to-event prediction arises in many domains, including healthcare, customer churn and component reliability, and is commonly addressed within *survival analysis* (Wang et al., 2019). A defining feature of survival data is *censoring*: for many individuals, the event time $T$ is not fully observed,

---
[1]SAFRAN Tech, Digital Sciences & Technologies Department, Rue des Jeunes Bois, Châteaufort, 78114 Magny-les-Hameaux, France [2]INSA Rennes, CNRS, IRMAR - UMR 6625, F-35000 Rennes, France [3]INRIA centre, Rennes university, Campus de Beaulieu, 263 Av. Général Leclerc, 35042 Rennes, France. Correspondence to: Ghanem Bahrini <ghanem.bahrini@safrangroup.com>, Ghanem Bahrini <ghanem.bahrini@insa-rennes.fr>.

*Proceedings of the 43rd International Conference on Machine Learning*, Seoul, South Korea. PMLR 306, 2026. Copyright 2026 by the author(s).

and a censored time is available (Commenges, 1998). This distinguishes survival analysis from standard regression and classification, as censoring affects not only model learning but also the evaluation of predictive performance. Importantly, both censoring mechanisms and censoring rates vary substantially across studies and application domains (Commenges, 1998; Wang et al., 2019).

Censoring is commonly categorized as *non-informative* or *informative*. A frequent source of *informative* censoring is *covariate-dependent censoring*, where the censoring distribution depends explicitly on observed covariates. When these covariates also predict the event process, this dependence induces informative censoring (Robins & Rotnitzky, 1992; Hernán et al., 2004). By contrast, *non-informative* censoring includes *independent censoring*, when censoring times are independent of the event times (conditional on covariates) and provides a canonical baseline (Prentice, 1978), as well as *administrative censoring*, a common special case arising from study-end truncation: individuals may enter the study at different times but are censored at a common study close if the event has not occurred (Commenges, 1998; Kvamme & Borgan, 2023). Throughout, we refer to administrative and independent censoring as *non-informative*, and to covariate-dependent censoring as *informative*.

In survival analysis, model evaluation is challenging due to the probabilistic nature of time-to-event predictions and the frequent presence of censoring. As a result, several complementary families of evaluation metrics have been proposed, each capturing a distinct aspect of predictive performance. Concordance-based measures, most notably the concordance index (C-index) (Harrell Jr et al., 1996; Uno et al., 2007), assess discriminative ability by ranking individuals by relative risk, but do not evaluate the accuracy of predicted survival probabilities. In contrast, probability-based metrics such as the Integrated Brier Score (IBS) (Graf et al., 1999; Gerds & Schumacher, 2006) evaluate the accuracy of predicted survival distributions over time, combining discrimination and calibration. Finally, calibration-focused criteria, including D-calibration (D'Agostino & Nam, 2003; Haider et al., 2020; Davidov et al., 2025), assess whether predicted survival probabilities are statistically consistent with observed event times at the population level. Taken together, these metric families provide a multidimensional view of performance encompassing discrimination, calibration, and

temporal accuracy—and underscore that appropriate criteria depend on the censoring mechanism and application setting (Lillelund et al., 2025b).

Although survival metrics capture complementary aspects of model performance, their validity critically depends on censoring assumptions. In particular, recent work has shown that widely used criteria can lead to misleading conclusions under informative or dependent censoring and has called for more principled evaluation practices (Lillelund et al., 2025a). Beyond this general observation, intrinsic sources of bias inherent to concordance-based measures have been analyzed (Alabdallah et al., 2024). Another work has further demonstrated that the C-index and the Brier score become biased when event and censoring times are statistically dependent (Gharari et al., 2023). Finally, prior work has shown that standard IPCW-based Brier score estimators are sensitive to the censoring mechanism, being biased under administrative censoring (Kvamme & Borgan, 2023) as well as under censoring dependent on covariates when the censoring model is misspecified (Prince et al., 2025).

Despite these known limitations, censoring-aware metrics are often treated as robust by default, with Harrell's and Antolini's C-index and IPCW-based IBS still dominating the evaluation of modern survival models (Wiegrebe et al., 2024). Model performance is typically summarized through fold-averaged scores (e.g., mean $\pm$ standard deviation), without explicitly examining how the censoring rate or the censoring mechanism may still affect the resulting values. This evaluation paradigm is pervasive across modern survival models, including deep learning approaches (Lee et al., 2018; Fotso, 2018; Norman et al., 2024) as well as more recent tree-based or interpretable methods (Zhang et al., 2024). While convenient, such aggregate summaries implicitly assume that evaluation metrics remain reliable under censoring, although when this assumption fails, metric values may become numerically biased and lead to incorrect model selection or unstable model rankings.

However, despite the recent critics, the literature still lacks a systematic analysis that disentangles (i) numerical bias in metric values from errors in model ranking, and (ii) the respective roles of the censoring rate and the censoring mechanism in driving these evaluation distortions across families of survival metrics. To address this gap, we adopt a controlled experimental framework based on *semi-synthetic censoring generation*, in which we start from fully observed event times and artificially generate censoring according to predefined mechanisms (administrative, independent, covariate-dependent) and rates. This design allows us to retain access to the true event times for all individuals while observing their censored counterparts. As a result, each experiment admits a *double evaluation*: a standard evaluation based on censored observations and an oracle

evaluation based on fully observed event times. This controlled comparison enables a unified analysis of censoring effects, linking numerical distortions of metric values with their consequences for model ranking.

**Contributions.** Our main contributions are threefold:

- We introduce a controlled evaluation protocol (standard vs. oracle) that disentangles the effects of the censoring *mechanism* and *rate* across major families of survival metrics, and quantify the resulting bias.

- We provide a principled analysis of how censoring-induced distortions translate into *model ranking instability*, showing when and why standard Top-1 benchmarking practices can lead to unreliable "best-model" conclusions.

- Based on these findings, we derive practical guidelines for the appropriate use and interpretation of survival evaluation metrics under different censoring scenarios, highlighting when standard model comparisons become unreliable.

**Main findings.** Across datasets and metrics, distortions are primarily driven by the amount of censoring, while the censoring mechanism plays a secondary, metric-dependent role. While global model rankings are often largely preserved, we show that Top-1 instability can substantially overstate apparent ranking failures and lead to erroneous conclusions when statistical tests are not used to assess differences in model performance.

Code and data processing scripts will be made publicly available at: `https://github.com/Ghanem01/When-Can-We-Trust-Survival-Model-Evaluation.git`.

## 2. Methodology

### 2.1. Survival Data & Models

We consider a right-censored survival dataset[1] with $N$ observations : $\mathcal{D} = \{(x_i, y_i, \delta_i)\}_{i=1}^{N}$. For subject $i$, $x_i \in \mathbb{R}^d$ denotes the observed covariates, $y_i \in \mathbb{R}_+$ the observed time, and $\delta_i \in \{0, 1\}$ the event indicator. Specifically, $\delta_i = 1$ indicates that the event occurred at time $y_i$, whereas $\delta_i = 0$ indicates right-censoring at time $y_i$. Each subject is associated with an event time $T_i$ and a censoring time $C_i$ such as $y_i = \min(T_i, C_i)$.

Given this censored data structure, we focus on *individual survival distribution* (ISD) models, which estimate, for each

---

[1]Right-censoring is the most common censoring setting in practice, where the event of interest has not yet occurred by the end of follow-up (Commenges, 1998).

subject $i$, the conditional survival function $S(t \mid x_i) = \mathbb{P}(T_i > t \mid X = x_i)$. This formulation underlies the evaluation framework considered in this work.

## 2.2. Tested Evaluation Metrics

**Concordance-based metrics.** The concordance index (C-index) measures a model's discriminative ability by quantifying the proportion of correctly ordered *comparable* pairs, where $(i, j)$ is comparable if $\delta_i = 1$ and $y_i < y_j$. Harrell's C-index (Harrell Jr et al., 1996) is the most widely used variant and relies on a global risk score $\hat{r}_i$ (defined in Appendix A.1),[2] yielding

$$C_{\text{Harrell}} = \frac{\sum_{i \neq j} \mathbf{I}\{y_i < y_j, \delta_i = 1\} \mathbf{I}\{\hat{r}_i > \hat{r}_j\}}{\sum_{i,j} \mathbf{I}\{y_i < y_j, \delta_i = 1\}}. \quad (1)$$

Antolini's C-index (Antolini et al., 2005) accounts for time-dependent predictions by assessing concordance using survival probabilities evaluated at the event time of the case, i.e., by comparing $\hat{S}(y_i \mid x_i)$ and $\hat{S}(y_i \mid x_j)$ for each comparable pair.[3] We additionally evaluate concordance at fixed horizons $C(t_q)$, with $t_q$ set to the 25%, 50%, and 75% quantiles of the training event-time distribution, contrasting individuals who experience the event before $t_q$ with those who remain event-free beyond $t_q$. This criterion corresponds to the cumulative/dynamic time-dependent concordance (AUC) (Heagerty et al., 2000; Heagerty & Zheng, 2005) and can be viewed as a fixed-horizon variant of the C-index. In addition, we study inverse probability of censoring weighted (IPCW) variants of both Harrell's and Antolini's C-indices. Formal definitions and implementation details are provided in Appendix A.1.

**Integrated Brier score.** The Integrated Brier Score (IBS) evaluates the accuracy of predicted survival distributions by integrating the Brier score over time. For a fixed time $t$, the (uncensored) Brier score is

$$\text{BS}(t) = \mathbb{E}\big[(\mathbf{I}\{T > t\} - \hat{S}(t \mid X))^2\big].$$

Because $\mathbf{I}\{T > t\}$ is unobserved for right-censored individuals, a standard correction relies on inverse probability of censoring weighting (IPCW) (Gerds & Schumacher, 2006), yielding the estimator

$$\text{BS}_{\text{IPCW}}(t) = \frac{1}{N} \sum_{i=1}^{N} \Bigg[ \frac{\mathbf{I}\{T_i \leq t, \delta_i = 1\}}{\hat{G}(T_i)} \hat{S}(t \mid x_i)^2$$
$$+ \frac{\mathbf{I}\{y_i > t\}}{\hat{G}(t)} \big(1 - \hat{S}(t \mid x_i)\big)^2 \Bigg], \quad (2)$$

---

[2]Harrell's C-index is computed using the `lifelines` implementation.

[3]Antolini's C-index is computed using the `pycox` implementation.

where $\hat{G}(t) = \mathbb{P}(C_i > t)$ denotes the Kaplan–Meier estimate of the censoring survival function. The IBS is obtained by integrating $\text{BS}_{\text{IPCW}}(t)$ over $\mathcal{T}_{\max} = [0, t_{\max}]$, where $t_{\max}$ is the maximum observed time on the evaluation grid.

Under heavy censoring, IPCW weights can become unstable as the estimated censoring survival function $\hat{G}(t)$ approaches zero. Standard remedies include weight clipping or restricting evaluation to earlier time horizons (Kvamme & Borgan, 2023). In this study, we adopt the latter strategy and report truncated variants $\text{IBS}(t_{25})$, $\text{IBS}(t_{50})$, and $\text{IBS}(t_{75})$, where $t_q$ denotes the corresponding quantile of the event-time distribution. As an alternative to IPCW, we consider a pseudo-observation–based estimator that replaces $\mathbf{I}\{T_i > t\}$ by leave-one-out Kaplan–Meier–based quantities (Cortese et al., 2013; Spitoni et al., 2018), avoiding explicit inverse weighting and providing a distinct correction for censoring. Finally, under *administrative censoring*, the IBS admits a dedicated estimator that exploits the known administrative censoring time for all individuals (Kvamme & Borgan, 2023). This variant is only defined in this setting and is evaluated separately. Formal definitions and implementation details are provided in Appendix A.2.

**Calibration metrics.** To assess the calibration of predicted survival distributions, we consider *D-Calibration* and *1-Calibration* (Haider et al., 2020). D-Calibration evaluates whether predicted survival probabilities at the *true event times* are uniformly distributed over $[0, 1]$. Partitioning $[0, 1]$ into $K = 10$ equal bins and letting $p_k$ denote the empirical proportion of probability mass in bin $k$, we quantify miscalibration, rather than using the original $\chi^2$ test—by the $L^1$ deviation from uniformity, $\text{D-Calib} = \sum_{k=1}^{K} \big|p_k - \frac{1}{K}\big|$, where smaller values indicate better calibration.

Complementarily, *1-Calibration* assesses calibration at a fixed time horizon $t^*$ by comparing predicted and observed event frequencies within risk groups using a Hosmer–Lemeshow–type statistic. We report 1-Calibration at the 25%, 50%, and 75% quantiles of the event-time distribution. Implementation details are provided in Appendix A.4.

## 2.3. Experimental Setup

*Table 1.* Summary of the five survival datasets used. Columns: (1) initial censoring rate, (2) number of subjects, (3) number of features, (4) maximum observed event time.

| DATASET | % CENS. | # SUBJ. | # FEAT. | MAX $t$ |
|---|---|---|---|---|
| NACD | 36.5 | 2,395 | 48 | 84.3 |
| GBMLGG | 44.2 | 1,102 | 13 | 6,423 |
| METABRIC | 55.2 | 1,981 | 79 | 9,218 |
| PBC | 61.5 | 418 | 17 | 4,795 |
| FLCHAIN | 72.5 | 7,871 | 23 | 5,215 |

We consider five publicly available survival datasets (Ta-

ble 1; Appendix B) and construct semi-synthetic versions with *controlled censoring*, varying both the censoring rate and mechanism while retaining access to event and censoring times. For each dataset, we generate target censoring rates $\rho \in \{10, 30, 50, 70, 90\}\%$ under three mechanisms—administrative (AC), independent (IC), and covariate-dependent (CDC) with 5 independent replications per configuration (dataset $d$, mechanism $c$, rate $\rho$) to account for stochastic variability.

**Controlled semi-synthetic censoring generation.** Our semi-synthetic construction produces datasets with a prescribed censoring rate while retaining, for each individual, both the true event time $T_i$ and a generated censoring time $C_i$. This enables paired evaluations based on standard censored observations and oracle evaluations. For CDC, individual-specific censoring probabilities are estimated using a Cox model for the censoring process. For IC, censoring probabilities are derived from a marginal Kaplan–Meier estimate and are therefore identical across individuals, removing any dependence on covariates. For AC, censoring is induced through study-end truncation, corresponding to a fixed calendar-time study close combined with entry-time–dependent follow-up. Full details of the generation procedures, including subject selection, censoring-time assignment, and rate matching, are provided in Appendix C.

On each generated dataset, we train a set of classical and modern survival models that estimate individual survival distributions. Specifically, we consider Cox proportional hazards (CoxPH) (Cox, 1972), Weibull accelerated failure time models (Kalbfleisch & Prentice, 2002), Random Survival Forests (RSF) (Ishwaran et al., 2008), and neural ISD models including DeepSurv (Katzman et al., 2018), DeepHit (Lee et al., 2018), Neural Multi-Task Logistic Regression (NMTLR) (Fotso, 2018), and Deep Cox Mixtures (DCM) (Nagpal et al., 2021). For each model, we compute the survival predictions required by the evaluation metrics considered in this study.[4]

**Models and training protocol.** For neural ISD models, hyperparameters are tuned via nested cross-validation using `Optuna` (3 inner folds / 5 outer folds). Crucially, we *retain* fold-level outputs: For each $(d, c, \rho)$, replication, and model, we store the metric values on each of the five outer test folds. To keep the training protocol fixed across metrics, since our goal is to study *metric behavior under censoring* rather than to maximize performance for a specific criterion, we perform hyperparameter selection using C-HARRELL. All implementation details and search spaces are provided in Appendix D.

---

[4]CoxPH and Weibull AFT are implemented using `lifelines`, RSF using `scikit-survival`, and Deep-Surv, DeepHit, NMTLR, and Deep Cox Mixtures using `pycox`.

**Standard (ST) vs. oracle (OR) evaluation.** For any metric $m$, each semi-synthetic dataset admits two evaluations, denoted ST and OR. In ST, $m$ is computed from the right-censored test data $\{(x_i, y_i, \delta_i)\}$, as in conventional practice. In OR, we compute the same metric after replacing each generated censoring time with its true event time, i.e., we use $(x_i, T_i, 1)$ instead of $(x_i, y_i, 0)$, yielding a reference that isolates the distortion due to censoring in the *metric computation* itself. This paired ST/OR design is applied uniformly across all $(d, c, \rho)$ settings and replications. Importantly, the OR evaluation is never observable in real applications and is used here solely as a controlled reference to isolate censoring-induced distortions.

## 2.4. Bias and Ranking Preservation Criteria

We quantify (i) how much censoring biases survival metrics compared to an oracle evaluation, and (ii) whether this bias translates into unreliable *model selection* and *model ranking*.

**Normalized metric bias.** For a metric $m$, and a configuration $(d, c, \rho)$, we denote by $m_r^{\mathrm{ST}}$ and $m_r^{\mathrm{OR}}$ the *standard* and *oracle* metric values for a given realization $r$ (model, fold, replication). We define the oracle deviation

$$\Delta m_r(d, c, \rho) = m_r^{\mathrm{ST}}(d, c, \rho) - m_r^{\mathrm{OR}}(d, c, \rho). \quad (3)$$

To make errors comparable across metrics and datasets, we normalize $\Delta m_r$ by a robust scale estimate of the oracle metric dispersion at the same $(d, c, \rho)$ level. Let $\mathcal{R}(d, c, \rho)$ be the set of all available realizations (all models, folds, and replications) under $(d, c, \rho)$, and let

$$\mathrm{IQR}\big(m^{\mathrm{OR}}(d, c, \rho)\big) = Q_{0.75}\big(\{m_r^{\mathrm{OR}}(d, c, \rho)\}_{r \in \mathcal{R}(d,c,\rho)}\big) \\ - Q_{0.25}\big(\{m_r^{\mathrm{OR}}(d, c, \rho)\}_{r \in \mathcal{R}(d,c,\rho)}\big), \quad (4)$$

where $Q_q$ denotes the empirical $q$-quantile. We then define the *normalized bias* for each realization as

$$B_r(d, c, \rho) = \frac{\Delta m_r(d, c, \rho)}{\mathrm{IQR}(m^{\mathrm{OR}}(d, c, \rho)) + \varepsilon}, \quad (5)$$

with a small $\varepsilon > 0$ used only for numerical stability. We use the interquartile range because it provides a *robust* proxy for the *natural dispersion* of the oracle metric (less sensitive to outliers than the standard deviation), while remaining directly comparable across heterogeneous metrics. In particular, $B_r(d, c, \rho) > 1$ indicates that the standard–oracle deviation exceeds the oracle IQR at the same setting, i.e., the bias is larger than the robust measure of intrinsic oracle variability.

**Ranking preservation: Top-1 consistency.** Beyond numerical bias, we assess whether distortions induced by censoring translate into *incorrect model selection* compared to the oracle evaluation. For each $(d, c, \rho)$ and each replication, we first compute a *fold-pooled* score (mean across

folds) for every model, independently for the standard and oracle evaluations. We then assess whether the model selected as better under the ST evaluation coincides with the OR-selected model.

Formally, Top-1 consistency is defined as

$$\mathbb{I}\left(\underset{m}{\text{opt}}\ \bar{s}_m^{\text{ST}}\ =\ \underset{m}{\text{opt}}\ \bar{s}_m^{\text{OR}}\right), \qquad (6)$$

where $\text{opt} \in \{\arg\max, \arg\min\}$ depends on whether the metric is to be maximized or minimized, and $\bar{s}_m$ denotes the fold-pooled score of model $m$. Averaging this indicator over replications and datasets yields the Top-1 accuracy reported in the results, together with uncertainty estimates based on the standard error of the mean (SEM).

**Global ranking agreement on comparable model pairs.** We additionally quantify agreement of the *entire* model ordering. For each $(d, c, \rho)$ and replication, we compare every pair of models $(i, j)$ using fold-aligned metric differences $\Delta_k = m_{i,k} - m_{j,k}$ for $k = 1, \ldots, 5$. Because fold-level estimates obtained under cross-validation are not independent, we use the cross-validation-corrected paired $t$-test (Bengio & Grandvalet, 2004), with corrected variance

$$\text{Var}_{\text{corr}} = \left(\frac{1}{n} + \frac{n_{\text{test}}}{n_{\text{train}}}\right) s^2, \qquad (7)$$

where $s^2$ is the empirical variance of the fold-wise paired differences, $n$ is the number of folds, and $n_{\text{train}}$ and $n_{\text{test}}$ denote the training and test sizes within each cross-validation split.

We run the test separately under the oracle and the standard evaluations. A pairwise comparison is retained only if it is significant under *both* evaluations ($p < 0.05$). If significant and the dominance direction agrees between oracle and standard, the pair is counted as *concordant* ($C$); if the directions disagree, it is counted as *discordant* ($D$). Otherwise, the pair is ignored. For each $(c, \rho)$, we aggregate $C$ and $D$ over datasets and replications and compute the dominance-based agreement score

$$\tau_{\text{dom}}(c, \rho)\ =\ \frac{C(c, \rho) - D(c, \rho)}{C(c, \rho) + D(c, \rho)}. \qquad (8)$$

This Kendall-like coefficient compares fold-level *groups* rather than single point estimates and discards non-significant pairs, making the agreement assessment more robust to noise. We additionally report the *coverage* ratio $(C + D)/(\text{total model pairs})$.

## 3. Results

### 3.1. Metric Bias Under Controlled Censoring

**Global bias patterns.** Figure 1 aggregates results across all five datasets, as the same qualitative trends are con-

sistently observed; per-dataset results are reported in Appendix E.3.1.

A first-order effect in Figure 1 is the censoring rate: across metric families and mechanisms, normalized bias generally increases with $\rho$. This behavior admits a direct methodological explanation. For concordance-based metrics, increasing censoring reduces the number of comparable pairs in the standard evaluation, whereas the oracle exploits all pairs. For Brier-score–based metrics, censored individuals contribute only up to their censoring time, effectively truncating the integration horizon in the standard setting. Calibration metrics exhibit a related but distinct effect: censored individuals contribute via redistribution mechanisms that spread probability mass across bins (Appendix A.4); as censoring increases, a growing fraction of individuals is incorporated in a near-uniform manner, which can artificially improve apparent calibration under standard evaluation. In contrast, oracle calibration relies exclusively on fully observed event times and thus reflects the true alignment between predicted and empirical risks. Beyond this global trend, bias magnitude also depends on the censoring mechanism in a metric-dependent manner: some criteria remain relatively stable under AC/IC, whereas others exhibit substantial distortions even under non-informative censoring.

**Concordance-family.** Within the C-index family, standard evaluation is predominantly biased in a *positive* direction, often *overestimating* oracle concordance and thus yielding an overly optimistic view of discrimination under heavy censoring. We observe a clear split between the global concordance measures (C-Harrell, C-Antolini, and C-Uno) and the fixed-horizon variants $C(t_q)$. Across censoring mechanisms, C-Harrell, C-Antolini, and C-Uno are consistently more biased than $C(t_q)$, with a magnitude that strongly depends on both the censoring mechanism and rate. C-Uno, which corresponds to the KM-based IPCW version of C-Harrell, exhibits bias patterns that closely follow those of C-Harrell and C-Antolini. Additional IPCW-corrected variants, including the Cox-based version of C-Uno and the KM- and Cox-based IPCW versions of C-Antolini, are reported in Appendix E.1; metric definitions are given in Appendix A.1. Under IC, substantial bias typically appears only at high censoring levels ($\rho \geq 70\%$), whereas under AC and especially under CDC, it can already become noticeable at moderate censoring levels ($\rho \geq 50\%$). As censoring increases, the set of comparable pairs both shrinks and shifts in composition: the proportion of event–event pairs decreases, and concordance becomes increasingly driven by comparisons involving censored observations (Alabdallah et al., 2024). In this regime, models may appear performant by exploiting censoring-related patterns, a failure mode that is strongest under informative censoring (CDC), rather than by correctly ordering true event times. In contrast, the $C(t_q)$ variants remain close to the oracle over a wide range of censoring lev-

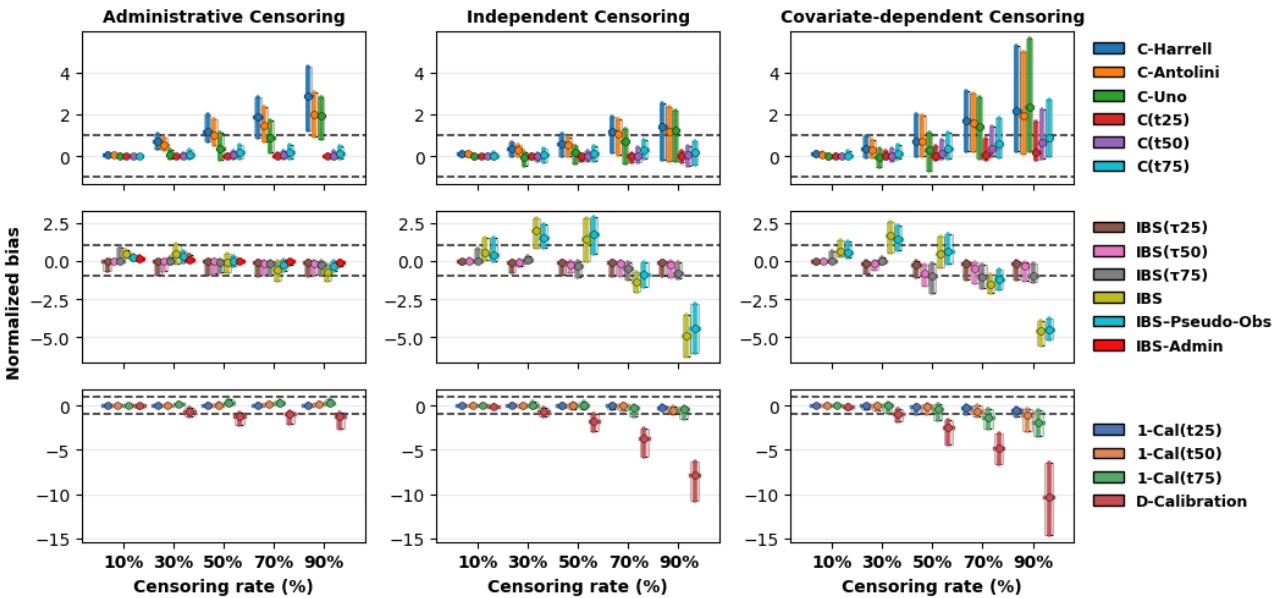

*Figure 1.* **Normalized Bias** (Eq. (5)) of survival evaluation metrics under controlled censoring. Boxplots aggregate results across all datasets, models, folds, and replications, and are shown for increasing censoring rates under three censoring mechanisms: administrative, independent, and covariate-dependent. Dashed horizontal lines indicate normalized bias values of $+1$ and $-1$, corresponding to deviations of one oracle interquartile range.

els, except under very strong CDC. Bias further decreases as the evaluation horizon shortens, with $C(t_{25})$ being the most robust, since fewer individuals are censored before $t_q$, reducing the impact of censoring on the metric.

**IBS-family.** Under AC, all IBS variants exhibit low normalized bias, with the mechanism-specific IBS-ADMIN being the most stable by construction. This behavior follows from the interaction between administrative truncation and the event-time distribution. As the censoring rate increases, administrative censoring mainly removes individuals with large event times, so that only a small fraction of censored subjects have true event times within the effective integration horizon (defined by observed training events). Consequently, censored individuals contribute almost identically under ST and OR evaluations over this horizon, yielding only minor differences in IBS. As a result, even at high censoring rates, IBS-type criteria remain close to their oracle counterparts under AC. The mechanism-specific IBS-ADMIN is the most stable, as it is tailored to administrative censoring and explicitly accounts for the study-end structure (Kvamme & Borgan, 2023).

Under both IC and CDC, IBS and IBS-PSEUDO-OBS display very similar bias patterns across censoring rates. The bias changes sign as $\rho$ increases: at low-to-moderate $\rho$, the standard score is slightly *worse* than the oracle (positive bias), whereas at high $\rho$ it becomes *overly optimistic* (negative bias). Our interpretation is that, at low-to-moderate

censoring, standard evaluation removes part of the event-time information while the effective integration window remains sufficiently broad, which can make the estimated prediction error slightly larger than under the oracle evaluation. At high censoring, this behavior reverses: the effective estimable horizon contracts, and the IBS becomes increasingly dominated by earlier time regions, where prediction errors are typically smaller, leading to artificially optimistic standard scores relative to the oracle. The fact that this behavior appears for both IPCW and pseudo-observation estimators indicates that it is not tied to a specific correction scheme or censoring mechanism, but rather reflects a progressive loss of temporal identifiability under heavy censoring. We emphasize that this explanation should be viewed as an empirical intuition rather than a formal characterization of IBS bias; a deeper theoretical analysis of this sign change remains an interesting direction for future work. Under IC, robustness is improved by restricting the integration horizon. Truncated variants (IBS($t_{25}$), IBS($t_{50}$), IBS($t_{75}$)) are consistently less biased than full-horizon IBS and IBS-PSEUDO-OBS, with IBS($t_{25}$) typically achieving the smallest bias. This is expected, as heavy censoring makes late-time contributions poorly supported: IPCW weights, Kaplan–Meier estimates, and pseudo-outcomes become highly variable in the tail, and truncation limits their influence. Cox-based IPCW IBS variants (Appendix E.2) behave similarly to the KM-based versions, with negligible differences for truncated horizons and slightly larger bias

*and variance* for full-horizon IBS.

In contrast, under CDC, all IBS variants become substantially biased once censoring is moderate-to-high (notably for $\rho \geq 50\%$). Importantly, this degradation is not limited to full-horizon criteria: even truncated IBS variants can exhibit large bias at high $\rho$, indicating that informative censoring broadly affects the IBS family and cannot be fully mitigated by horizon restriction.

**Calibration-family.** While normalized bias remains limited under administrative censoring, D-CALIB deteriorates sharply under both independent and covariate-dependent censoring, with a pronounced (often non-linear) increase as the censoring rate grows. Importantly, the bias is consistently *negative*: ST systematically suggests *better calibration* than OR, thereby understating miscalibration across censoring rates and mechanisms, most notably for D-CALIB. This behavior follows directly from how censoring is incorporated in D-CALIB (Haider et al., 2020). Under heavy censoring, many individuals are censored early, so the predicted survival at the censoring time, $S_i(C_i)$, is often close to 1. By construction, censored observations contribute by *spreading* probability mass over all bins below $S_i(C_i)$; when $S_i(C_i) \approx 1$, this mass is distributed almost uniformly over $[0, 1]$. As early censoring becomes frequent, these near-uniform contributions can dominate bin counts and artificially equalize them, creating an *illusion of good calibration* even for misspecified models. This effect is much weaker under AC, where censoring occurs late in follow-up and censored individuals contribute less to early-horizon calibration. Supporting distributional evidence is reported in Appendix C.4.

In contrast, 1-Calibration metrics remain comparatively more stable. Their normalized bias stays limited at low to moderate censoring levels and only becomes pronounced at high censoring rates ($\rho = 70\%, 90\%$) when the censoring mechanism is covariate-dependent. This relative robustness can be attributed to the way censoring is handled in 1-CAL. At a fixed evaluation horizon $t^*$, calibration is assessed *locally* within bins of predicted risk: for each bin, the metric contrasts the model-implied event probability by $t^*$ with an event probability estimated from the right-censored outcomes using a bin-specific Kaplan–Meier correction.[5] Unlike D-CALIB, this procedure does not redistribute each censored individual's mass across the full $[0, 1]$ range, which limits the kind of global "smoothing" effect that can mask miscalibration. Empirically, the signed bias is predominantly negative, indicating that censoring tends to make calibration appear *better* than under OR (i.e., ST overstates

---

[5]1-Calibration evaluates calibration at a fixed $t^*$ within predicted-risk bins by contrasting the model-implied event probability with a bin-wise Kaplan–Meier estimate from the right-censored data.

calibration), even though 1-CAL typically degrades more gradually than D-CALIB as $\rho$ increases.

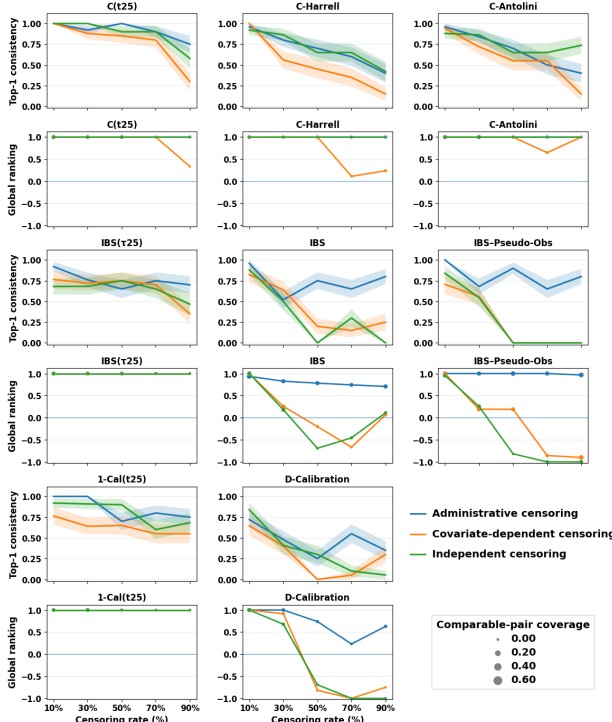

*Figure 2.* **Preservation of model ranking under censoring.** The figure contrasts Top-1 consistency (odd rows) with a dominance-based global ranking agreement score (even rows), computed from statistically comparable model pairs, across censoring rates and censoring mechanisms. Corresponding figures for the other evaluation metrics are reported in Appendix E.3.2.

### 3.2. Ranking Preservation Under Censoring.

While relative bias quantifies numerical deviation from the oracle, it does not directly assess whether metrics preserve the *relative ordering of models*, which is often the primary objective in practice. Crucially, limited numerical bias does not imply robust model selection: even metrics that remain close to their oracle values could yield unstable Top-1 decisions once censoring increases.

**Top-1 consistency.** Top-1 consistency generally decreases with the censoring rate (Figure 2, odd rows), showing that the standard evaluation can select a different best model than the oracle. However, this raw Top-1 comparison should not be interpreted as evidence that the identity of a uniquely best model always changes: Top-1 decisions are sensitive to small differences between the two leading models, which may not be statistically distinguishable. This instability is strongly metric-dependent. Among concordance criteria, the widely used C-HARRELL and C-ANTOLINI exhibit a

continuous decrease, whereas time-dependent concordance at early horizons (e.g., $C(t_{25})$) is typically more resilient and degrades mainly at extreme informative censoring. For IBS family, IBS and IBS-PSEUDO-OBS lose Top-1 stability rapidly under independent and covariate-dependent censoring, while restricting the integration horizon (e.g., $IBS(t_{25})$) leads to more stable rankings, yet residual bias remains. For calibration, D-CALIB is the most unstable criterion: Top-1 consistency can fall sharply even at moderate censoring, and although early-horizon 1-CAL (e.g., $1\text{-}CAL(t_{25})$) degrades more slowly, it remains far from robust when censoring increases. More generally, while strong metric bias systematically leads to severe Top-1 instability across censoring mechanisms, we also observe that even settings with limited numerical bias—such as $IBS(t_{25})$ or $1\text{-}CAL(t_{25})$ under AC and IC—can induce non-negligible Top-1 instability. This highlights that small average distortions in metric values may still suffice to alter apparent best-model conclusions. To separate genuine Top-1 changes from statistically weak model separations, we further considered a significance-aware Top-1 analysis based on the same test used for global ranking. For each configuration, we retained a Top-1 comparison only when the gap between the top-ranked and second-ranked models was statistically significant under both oracle and standard evaluations. This filtering substantially reduces the number of comparable cases (from 225 to 130), especially for some calibration metrics and under heavier censoring. This indicates that in many settings no reliable winner can be identified. Thus, the issue is not only that censoring may change which model appears best, but also that, as censoring increases, the evidence for declaring a unique best model often becomes too weak.

Overall, these patterns suggest that best-model claims based on standard fold-averaged metrics should be interpreted cautiously under censoring, especially when differences between leading models are small.

**Global agreement.** Robust global ranking tells a different story (Figure 2, even rows) because the objective is different: rather than focusing on Top-1 alone, we test whether the *entire ordering* is preserved on model pairs that are *statistically different* under both oracle and standard evaluations (cf. Eq. (8)). The dominance-based agreement, unlike Top-1 consistency, deliberately ignores model pairs that are not statistically distinguishable. Since the coverage $(C + D)$ is often low, only a limited subset of model relations contributes to the score; in particular, the pair involving the top two models is often not statistically significant and therefore excluded. On these comparable pairs, several metrics yield near-perfect agreement ($\tau_{\text{dom}} \approx 1$) across a wide range of censoring settings. In particular, C-INDEX–based metrics, except when censoring is informative, can preserve a high global agreement even when Top-1 consistency deteriorates.

In contrast, IBS, IBS-PSEUDO-OBS, and D-CALIB can exhibit substantial agreement loss under (CDC) and (IC), indicating genuine rank inversions among discriminable pairs, whereas $IBS(t_{25})$ and $1\text{-}CAL(t_{25})$ recover near-perfect global agreement across most censoring mechanisms, despite their sensitivity to censoring in absolute bias.

## 4. Discussion

**Positioning w.r.t. recent evaluation critiques.** The evaluation of survival models under censoring has recently drawn increased attention, with several works questioning standard benchmarking practices by highlighting (i) intrinsic limitations of concordance-based criteria and pair comparability effects (Alabdallah et al., 2024), (ii) bias of Brier-score–based estimators under AC or CDC (Kvamme & Borgan, 2023; Prince et al., 2025), and (iii) failures of commonly reported calibration tests in the presence of censoring (Gharari et al., 2023; Lillelund et al., 2025a). However, existing evidence remains fragmented, as results are often reported for a single metric family or censoring regime without disentangling *numerical bias* from *model ranking reliability*. Our controlled STD/ORC protocol addresses this gap by explicitly separating numerical metric bias from ranking distortions and isolating the respective roles of censoring rate and mechanism, thereby clarifying long-standing ambiguities in survival benchmarking practice.

**Censoring rate dominates, but mechanisms determine failure modes.** Across all metric families, the censoring rate is the main driver of evaluation distortion: as censoring increases, standard evaluations depart from oracle assessments numerically and weaken the reliability of model-selection conclusions. Failure modes, however, depend critically on the censoring mechanism and the metric. Under non-informative censoring (AC/IC), global rankings for discrimination-oriented metrics are often preserved despite biased scores, whereas informative censoring (CDC) alters the effective information content and can induce genuine ranking inversions. This explains why metrics may appear robust in some settings but fail in others, depending on the censoring regime, evaluation horizon, and metric choice (Birolo et al., 2025; Kvamme & Borgan, 2023; Lillelund et al., 2025b). Empirically, under CDC, numerical bias and ranking distortions emerge already at moderate censoring levels ($\rho \approx 50\%$). In contrast, under AC and IC, numerical bias remains limited until high censoring rates ($\rho \geq 70\%$), while early Top-1 instability is largely driven by small, statistically weak differences between leading models; global ranking agreement degrades only at very high censoring levels and is largely confined to tail-dependent comparisons.

**Top-1 benchmarking requires statistical caution under censoring.** Even when global ordering is largely preserved,

small censoring-induced perturbations can change the identity of the apparent top-ranked model when performance differences are marginal. This instability appears across metrics and censoring mechanisms and is not prevented by low numerical bias alone, but it should be interpreted carefully because many Top-1 differences are not statistically separable. Consequently, selecting a single "best model" based solely on fold-averaged scores—without statistical testing or sensitivity analysis—can be unreliable in survival settings, especially when the leading models have close performance. Significance-aware Top-1 analyses clarify that the issue is not only that censoring may change which model appears best, but also that, under heavier censoring, the evidence for declaring any unique winner often becomes weak. In contrast, dominance-based agreement shows that, when restricted to statistically meaningful comparisons, several metrics maintain high consistency with oracle rankings even under moderate-to-high censoring.

**Metric-specific implications for practice.** Our findings directly inform metric choice. Global concordance measures (Harrell/Antolini) often overestimate discrimination under heavy censoring—especially under informative censoring—whereas fixed-horizon concordance is more robust by reducing reliance on late-time comparisons. Under AC/IC, these C-index variants can still support meaningful model comparison when ranking is assessed via statistically grounded pairwise tests, despite apparent Top-1 volatility and weak separability between leading models. For probability-based metrics, horizon truncation helps under IC but not under CDC, where IBS variants remain unreliable, including IPCW-IBS with covariate-conditional (Cox) censoring models (Appendix E.2). However, this mitigation should not be interpreted as uniformly preferable for model selection: shorter horizons can reduce bias by avoiding poorly supported late-time regions, but they may also reduce the number of informative comparisons and increase variance, especially for concordance-based metrics. We analyze this bias–variance trade-off across horizons in Appendix A.3. Calibration is most fragile: D-Calibration can appear artificially well-behaved at high censoring, while 1-Calibration degrades more gradually but is still sensitive to CDC. Overall, no single metric is universally reliable; evaluation should be tailored to the censoring structure and interpreted with explicit awareness of its limitations.

**Broader implications and limitations.** More broadly, our results show that censoring-aware estimators are not censoring-agnostic: even when theoretically justified, their finite-sample behavior can be dominated by censoring-induced information loss and redistribution effects. This impacts not only benchmarking, but also hyperparameter tuning, model selection, and claims of methodological superiority. Finally, our controlled semi-synthetic framework enables clean oracle comparisons but necessarily abstracts

away from additional real-world complexities, such as unobserved confounding, competing risks, or time-dependent covariates. Extending the analysis to these settings remains an important avenue for future research.

**Recommendations.** Based on our findings, we make three practical recommendations: (i) always report censoring rates and, when possible, characterize the censoring mechanism; (ii) avoid Top-1-only conclusions and complement them with statistically grounded pairwise or group-level comparisons, including checks that the leading models are actually separable; and (iii) consider early-horizon or truncated metrics when late-horizon evaluation becomes unreliable, while explicitly checking the induced bias–variance trade-off and whether the chosen horizon remains scientifically relevant. In particular, under suspected CDC or heavy censoring, practitioners should complement standard IPCW-style corrections with simple instability diagnostics—such as effective coverage of comparable pairs for concordance-based metrics, summaries of IPCW weight variability, and horizon-wise variance diagnostics—and consider truncation when these indicate unreliable late-time estimation. Ultimately, trustworthy survival evaluation requires not only censoring-aware metrics, but also censoring-aware interpretation.

## Impact Statement

This paper presents work whose goal is to advance the field of Machine Learning. There are many potential societal consequences of our work, none which we feel must be specifically highlighted here.

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

# A. Details of Survival Evaluation Metrics

## A.1. Concordance-Based Metrics

As discussed in Section 2.2, concordance-based metrics quantify discrimination by comparing pairs of individuals whose outcomes can be ordered from the observed right-censored data. We detail here the exact risk score used for Harrell's C-index in our implementation, together with the formal definitions of Antolini's C-index and fixed-horizon concordance $C(t)$.

**Risk score used for Harrell's C-index.** Harrell's C-index (Eq. (1) in the main paper) requires a *scalar* risk score $\hat{r}_i$ per individual. For models that output an estimated survival curve $\hat{S}(\cdot \mid x_i)$ on a discrete time grid $\{t_m\}_{m=1}^M$, we construct a cumulative-risk score as the discrete integral of the cumulative incidence:

$$\hat{r}_i \;=\; \int \left(1 - \hat{S}(t \mid x_i)\right) dt \;\approx\; \sum_{m=1}^{M} \left(1 - \hat{S}(t_m \mid x_i)\right) \Delta t_m, \tag{9}$$

where we set $t_0 = 0$ and $\Delta t_m = t_m - t_{m-1}$ for $m = 1, \ldots, M$.

This score is monotone with risk (larger $\hat{r}_i$ indicates higher risk) and is used as input to Harrell's concordance computation.

**Antolini's time-dependent C-index.** Antolini's C-index replaces the comparison of scalar risks by a comparison of survival probabilities evaluated at the event time of the *case* (Antolini et al., 2005). Let $(i, j)$ be a *comparable* pair if $\delta_i = 1$ and $y_i < y_j$. Antolini's concordance counts the pair as concordant if the individual who fails first has lower predicted survival at time $y_i$:

$$\hat{S}(y_i \mid x_i) < \hat{S}(y_i \mid x_j).$$

Formally, defining

$$c_{ij}^{\mathrm{A}} = \mathbf{I}\{\hat{S}(y_i \mid x_i) < \hat{S}(y_i \mid x_j)\} + \tfrac{1}{2}\mathbf{I}\{\hat{S}(y_i \mid x_i) = \hat{S}(y_i \mid x_j)\},$$

Antolini's C-index is

$$C_{\mathrm{Antolini}} = \frac{\sum_{i \neq j} \mathbf{I}\{\delta_i = 1,\, y_i < y_j\}\, c_{ij}^{\mathrm{A}}}{\sum_{i \neq j} \mathbf{I}\{\delta_i = 1,\, y_i < y_j\}}. \tag{10}$$

Unlike Harrell's C-index, this induces a *variable evaluation horizon* because the comparison time is $t = y_i$ and thus differs across pairs.

**Fixed-horizon concordance $C(t)$.** We additionally evaluate discrimination at fixed horizons $t = t_q$ (with $t_q$ the 25%, 50%, and 75% quantiles of the event-time distribution). At a given horizon $t$, we form *time-specific* comparable pairs that contrast individuals who experience the event by $t$ with those who remain event-free beyond $t$. Individuals censored before $t$ (i.e., $y_i \leq t$ and $\delta_i = 0$) do not form comparable pairs at $t$, as their event status by $t$ is not identifiable.

A pair $(i, j)$ is comparable at $t$ if either

$$(\delta_i = 1,\; y_i \leq t,\; y_j > t) \quad \text{or} \quad (\delta_j = 1,\; y_j \leq t,\; y_i > t).$$

Concordance is assessed using the predicted survival probability at $t$: the individual who fails by $t$ should have *lower* predicted survival at $t$. Defining

$$c_{ij}(t) = \mathbf{I}\{\hat{S}(t \mid x_i) < \hat{S}(t \mid x_j)\} + \tfrac{1}{2}\mathbf{I}\{\hat{S}(t \mid x_i) = \hat{S}(t \mid x_j)\},$$

and letting $\mathcal{P}(t)$ denote the set of comparable pairs at time $t$, we compute

$$C(t) = \frac{1}{|\mathcal{P}(t)|} \sum_{(i,j) \in \mathcal{P}(t)} c_{ij}(t). \tag{11}$$

Compared to Antolini's C-index, $C(t)$ uses a *fixed* comparison time, but a different notion of comparability (event-by-$t$ vs event-free-beyond-$t$), which can alter sensitivity to censoring through the effective set of usable pairs.

**IPCW C-index.** In addition to Harrell's concordance index, we consider the inverse probability of censoring weighted (IPCW) C-index proposed by Uno et al. (2007), which can be viewed as an IPCW-corrected version of Harrell's global concordance. This criterion reweights comparable pairs to account for right-censoring. Let $G(t) = \mathbb{P}(C_i > t)$ denote the censoring survival function. For a comparable pair $(i, j)$ with $\delta_i = 1$ and $y_i < y_j$, Uno's estimator assigns a weight proportional to $1/G(y_i)^2$, yielding

$$C_{\text{IPCW}} = \frac{\sum_{i \neq j} \mathbf{I}\{\delta_i = 1, \, y_i < y_j\} \, \frac{1}{G(y_i)^2} \, \mathbf{I}\{\hat{r}_i > \hat{r}_j\}}{\sum_{i \neq j} \mathbf{I}\{\delta_i = 1, \, y_i < y_j\} \, \frac{1}{G(y_i)^2}}. \tag{12}$$

The same IPCW principle is also applied to Antolini's time-dependent C-index, yielding an IPCW-corrected version of event-time concordance.

**Censoring-model variants.** For both IPCW–Harrell (Uno) and IPCW–Antolini, we estimate the censoring survival function $G$ using two approaches. A Kaplan–Meier estimator provides a marginal estimate of $G$ that does not account for covariates, while a Cox proportional hazards model estimates $G(t \mid X)$ conditionally on covariates, explicitly modeling the censoring mechanism. This results in KM-based and Cox-based IPCW variants for both global (Harrell/Uno) and time-dependent (Antolini) concordance, all of which are evaluated in parallel in our experiments.

**Implementation note.** When $\hat{S}(\cdot \mid x_i)$ is provided on a discrete grid, we use nearest-neighbor evaluation for concordance-based metrics, and linear interpolation for Brier/IBS variants when values are required at intermediate times.

## A.2. Integrated Brier Score Variants

As discussed in the main paper (Section 2.2), the Integrated Brier Score (IBS) evaluates the accuracy of predicted survival distributions by integrating a pointwise Brier score over time. We provide here the formal definitions of the IBS variants considered (IPCW, pseudo-observations, and the administrative estimator), together with the horizon truncation strategy used in our experiments.

**IPCW Brier score and truncated IBS.** Under right-censoring, the indicator $\mathbf{I}\{T_i > t\}$ is not observable for individuals censored before $t$. A standard correction is to use inverse probability of censoring weighting (IPCW) (Graf et al., 1999; Gerds & Schumacher, 2006), based on the censoring survival function $G(t) = \mathbb{P}(C_i > t)$ estimated via Kaplan–Meier on the training censoring indicators. The resulting pointwise estimator is (Eq. (2) in the main paper)

$$\widehat{\text{BS}}_{\text{IPCW}}(t) = \frac{1}{N} \sum_{i=1}^{N} \left[ \frac{\mathbf{I}\{y_i \leq t, \, \delta_i = 1\}}{\hat{G}(y_i)} \, \hat{S}(t \mid x_i)^2 + \frac{\mathbf{I}\{y_i > t\}}{\hat{G}(t)} \left(1 - \hat{S}(t \mid x_i)\right)^2 \right].$$

In heavy censoring regimes, IPCW can become numerically unstable because $\hat{G}(t)$ may approach zero at large times, leading to exploding weights. Common practical remedies include truncating the integration horizon and clipping extreme weights. In particular, under suspected CDC or heavy censoring, practitioners should complement standard IPCW-style corrections with diagnostics for weight instability (and consider truncation), as apparent "corrections" can otherwise amplify noise.

In this work, we mitigate this instability by restricting integration to fixed horizons and reporting truncated variants $\text{IBS}(t_{25})$, $\text{IBS}(t_{50})$, and $\text{IBS}(t_{75})$, where $t_q$ denotes the $q$-th quantile of the *training* event-time distribution. For each horizon $t_q$, we compute

$$\text{IBS}(t_q) = \frac{1}{t_q} \int_0^{t_q} \widehat{\text{BS}}_{\text{IPCW}}(t) \, dt,$$

where the integral is approximated numerically on the model's prediction grid.

In addition to the standard Kaplan–Meier–based IPCW estimator, we also compute Cox-based IPCW IBS variants, where the censoring survival function $G(t \mid x)$ is estimated conditionally on covariates using a Cox proportional hazards model. This allows us to assess whether modeling covariate-dependent censoring improves IBS behavior compared to the marginal Kaplan–Meier approach.

**Pseudo-observation IBS.**    As an alternative to inverse weighting, we consider the pseudo-observation approach, following Cortese et al. (2013); Spitoni et al. (2018). Let $\hat{S}_{\text{KM}}(t)$ denote the Kaplan–Meier estimate of the survival function of $T$ computed on the test sample, and $\hat{S}_{\text{KM}}^{(-i)}(t)$ the corresponding leave-one-out estimate obtained by removing individual $i$. The pseudo-observation for $\mathbf{I}\{T_i > t\}$ is defined as

$$J_i(t) = N\, \hat{S}_{\text{KM}}(t) - (N-1)\, \hat{S}_{\text{KM}}^{(-i)}(t).$$

Using the identity $(\mathbf{I}\{T_i > t\} - \hat{S}_i(t))^2 = \mathbf{I}\{T_i > t\}\,(1 - 2\hat{S}_i(t)) + \hat{S}_i(t)^2$, the pointwise Brier score is estimated as

$$\widehat{\text{BS}}_{\text{pseudo}}(t) = \frac{1}{N}\sum_{i=1}^{N}\left[ J_i(t)\left(1 - 2\hat{S}(t \mid x_i)\right) + \hat{S}(t \mid x_i)^2 \right].$$

The pseudo-observation IBS is then obtained by integrating $\widehat{\text{BS}}_{\text{pseudo}}(t)$ over the same time grid as the IPCW-based IBS, using identical normalization. In our implementation, $\hat{S}(t \mid x_i)$ is evaluated on the model prediction grid (with linear interpolation when needed), and all pseudo-observations are computed at these grid points.

**Administrative IBS.**    Under administrative censoring, each individual $i$ has a known administrative censoring time $c_i^\star$, available for all subjects (including those with $\delta_i = 1$). We follow the estimator of Kvamme & Borgan (2023), which evaluates the Brier score at time $t$ only on the administratively observable subset

$$\mathcal{A}(t) = \{i:\ c_i^\star \geq t\}.$$

The pointwise administrative Brier score is then

$$\widehat{\text{BS}}_{\text{admin}}(t) = \frac{1}{|\mathcal{A}(t)|}\sum_{i\in\mathcal{A}(t)}\left(\tilde{Y}_i(t) - \hat{S}(t \mid x_i)\right)^2,$$

where $\tilde{Y}_i(t)$ is identifiable on $\mathcal{A}(t)$ from the observed pair $(y_i, \delta_i)$ under the convention used in our experiments. The administrative IBS is obtained by integrating $\widehat{\text{BS}}_{\text{admin}}(t)$ over $[0, t_q]$, further restricted to $t \leq \max_i c_i^\star$ to ensure $|\mathcal{A}(t)| > 0$. In our implementation, this estimator requires the column `admin_censor_time` (providing $c_i^\star$ for all individuals) and is computed only for experiments with administrative censoring.

### A.3. Variance Analysis Across Evaluation Horizons

To assess whether horizon truncation provides a favorable bias–variance trade-off, we additionally analyzed the variance of standard metric values across evaluation horizons. For each dataset, censoring mechanism, model, metric family, horizon, and censoring rate, we computed the empirical variance of the standard metric values across folds and replications. We considered horizon-specific C-index metrics $C(t_{25})$, $C(t_{50})$, $C(t_{75})$ together with full-horizon C-Harrell; truncated IBS metrics $\text{IBS}(t_{25})$, $\text{IBS}(t_{50})$, $\text{IBS}(t_{75})$ together with full-horizon IBS; and horizon-specific 1-Calibration metrics $1\text{-}\text{CAL}(t_{25})$, $1\text{-}\text{CAL}(t_{50})$, and $1\text{-}\text{CAL}(t_{75})$. The resulting variances were then averaged over datasets for each model, censoring mechanism, censoring rate, and horizon, and are reported in Figure 3.

Overall, the trade-off is metric-family dependent. For 1-Calibration, shorter horizons generally reduce bias and often also exhibit lower variance. For horizon-specific C-index metrics, variance tends to increase at shorter horizons, consistent with the smaller number of comparable pairs. For IBS, the pattern is less monotonic: at low censoring, variance tends to decrease with the horizon, whereas at high censoring it can increase toward later horizons. These results support the use of truncation as a pragmatic mitigation strategy when late-horizon evaluation becomes unreliable, but not as a universally preferable choice for model selection. In practice, the choice of horizon should also reflect the scientific or clinical time scale of interest.

### A.4. Calibration Metrics

As introduced in the main paper (Section 2.2), calibration metrics assess whether predicted survival distributions are statistically consistent with observed outcomes. We detail here the exact formulations used for D-Calibration and 1-Calibration, together with their censoring corrections and the scalar summaries reported in our experiments.

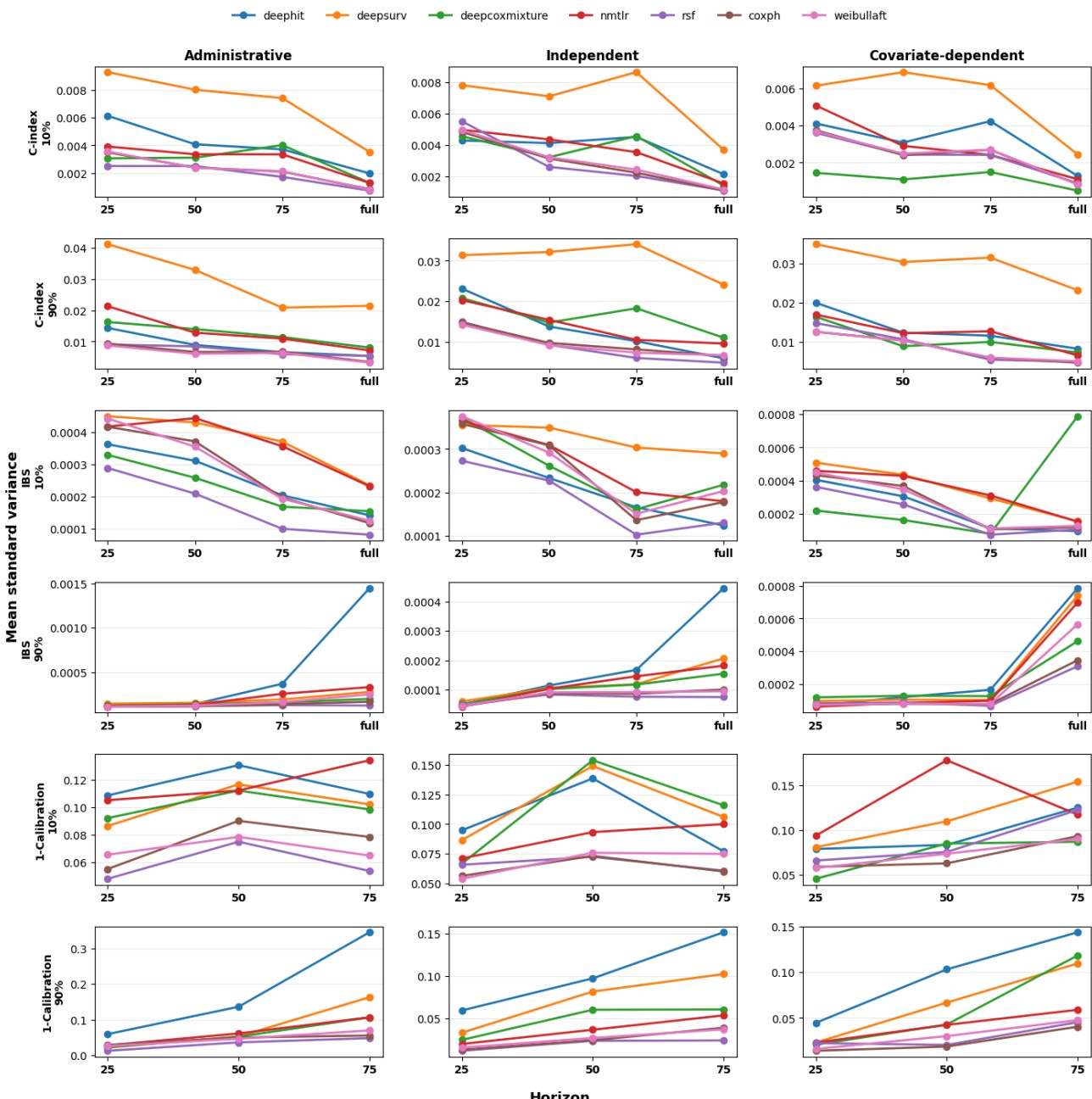

*Figure 3.* Mean variance of standard metric values across evaluation horizons. The $x$-axis shows the evaluation horizon for each metric family: $t_{25}$, $t_{50}$, $t_{75}$, and full horizon for C-index and IBS metrics, and $t_{25}$, $t_{50}$, and $t_{75}$ for 1-Calibration. The $y$-axis reports the mean variance of standard metric values averaged over datasets. Rows correspond to metric families and censoring rates, columns to censoring mechanisms, and curves to models.

**D-Calibration.** D-Calibration evaluates whether the predicted survival probabilities at the (unobserved) event times follow a uniform distribution on $[0, 1]$ (D'Agostino & Nam, 2003; Haider et al., 2020). In the absence of censoring, a well-calibrated model satisfies

$$\hat{S}(T_i \mid x_i) \sim \mathcal{U}[0, 1] \quad \text{for uncensored } i.$$

The unit interval is partitioned into $K$ equal-width bins; throughout this work we fix $K = 10$. Let $p_k$ denote the empirical proportion of mass assigned to bin $k$.

Under right-censoring, $\hat{S}(T_i \mid x_i)$ is not observable when $\delta_i = 0$. D-Calibration addresses this by distributing the contribution of censored individuals across bins according to the shape of their predicted survival curve beyond the censoring time (Haider et al., 2020). This yields a bin-wise empirical distribution $\{p_k\}_{k=1}^{K}$, which is normalized to sum to one.

Rather than reporting the $\chi^2$ test or its associated $p$-value, we summarize miscalibration using a scalar deviation-from-uniformity measure:

$$\text{D-Calib} = \sum_{k=1}^{K} \left| p_k - \tfrac{1}{K} \right|,$$

where smaller values indicate better calibration. This formulation directly quantifies the magnitude of miscalibration and is the version used throughout our experiments.

**1-Calibration (D'Agostino–Nam).** 1-Calibration evaluates calibration at a fixed time horizon $t^*$ by comparing predicted and observed event frequencies across risk bins (D'Agostino & Nam, 2003). Individuals are grouped into $K$ bins according to their predicted event probability $1 - \hat{S}(t^* \mid x_i)$, and $\bar{p}_j$ denotes the mean predicted event probability in bin $j$.

Direct comparison with observed events is biased under censoring. The D'Agostino–Nam correction addresses this by estimating, within each bin, a Kaplan–Meier curve using only individuals in that bin, yielding an estimate of the event probability at $t^*$:

$$\widehat{\Pr}(T \leq t^* \mid \text{bin } j) = 1 - \widehat{\text{KM}}_j(t^*).$$

Instead of reporting the associated Hosmer–Lemeshow $\chi^2$ statistic or its $p$-value, we summarize miscalibration at horizon $t^*$ by the absolute discrepancy between predicted and estimated event probabilities:

$$\text{1-Calib}(t^*) = \sum_{j=1}^{K} \left| \bar{p}_j - \left( 1 - \widehat{\text{KM}}_j(t^*) \right) \right|.$$

We report 1-Calibration at $t^* \in \{t_{25}, t_{50}, t_{75}\}$, corresponding to the 25%, 50%, and 75% quantiles of the event-time distribution.

**Limitations under heavy censoring.** Both D-Calibration and 1-Calibration rely on Kaplan–Meier–based corrections to handle censoring. When censoring is heavy or occurs early, these corrections may become overly permissive: censored individuals contribute diffuse or near-uniform mass across bins, and bin-specific Kaplan–Meier estimates may be based on few or no observed events. As a result, calibration metrics can appear artificially favorable even when predicted survival distributions are misspecified. This behavior motivates cautious interpretation of calibration scores under strong censoring, and complements the bias analyses reported in the main paper.

## B. Datasets and Preprocessing

This appendix provides additional details on the survival datasets used in our experiments and the preprocessing steps applied prior to censoring simulation. A summary of the datasets is reported in Table 1 of the main paper. Across all datasets, survival times are defined as the time from baseline to the event of interest, and right-censoring is handled using the provided event indicators. Unless stated otherwise, observations with non-positive survival times are removed. Categorical variables are encoded using either binary encoding or one-hot encoding, and continuous variables are kept on their original scale.

### B.1. GBMLGG (Glioma)

The *GBMLGG* dataset is derived from *The Cancer Genome Atlas* (TCGA) and contains survival data for patients diagnosed with glioma (Weinstein et al., 2013). Survival time is defined using the `days_to_death` variable, with censored observations completed using `days_to_last_follow_up`. The event indicator is derived from the vital status. Three variables

(radiation therapy status, Karnofsky performance score, and ethnicity) contain missing values, which are imputed using the median. Non-informative identifiers and redundant follow-up variables are removed. Gender, radiation therapy status, and ethnicity are encoded as binary variables, while race and histological subtype are one-hot encoded.

### B.2. METABRIC

The *METABRIC* dataset contains breast cancer survival data with clinical and molecular covariates, including age at diagnosis, tumor size, lymph node involvement, and the Nottingham Prognostic Index (Curtis et al., 2012). The dataset is accessed via the `pycox` repository. No missing values are present, and preprocessing is limited to the removal of observations with non-positive survival times.

### B.3. PBC (Primary Biliary Cholangitis)

The *Primary Biliary Cholangitis* dataset from the Mayo Clinic (Therneau, 1997) includes longitudinal clinical measurements for patients diagnosed with PBC. Survival time and event indicators are derived from the original status variables, with liver transplantation treated as a censoring event. Patient identifiers are removed. Continuous variables with missing values are imputed using their mean, and categorical variables using their mode. Treatment assignment is encoded as a binary variable.

### B.4. NACD (Northern Alberta Cancer Dataset)

The *NACD* dataset contains survival data for patients diagnosed with several cancer types, including lung, colorectal, and esophageal cancer (Haider et al., 2020). The event indicator is extracted from the `delta` variable. The dataset contains no missing values. Variables related to disease staging and performance status are excluded to focus on baseline covariates.

### B.5. FLCHAIN

The *FLCHAIN* dataset, available through `scikit-survival`, includes individuals whose serum free light chain levels (kappa and lambda) were measured (Dispenzieri et al., 2012). Survival time and event indicators are derived from the original variables. The ICD-9 classification column is removed. Missing values are imputed using the mean for continuous variables and the mode for categorical variables. Categorical features, such as sample year and free light chain group, are one-hot encoded.

## C. Synthetic Data Generation

To study the impact of censoring under controlled conditions, we construct semi-synthetic survival datasets that preserve the covariate structure and event-time distribution of real data while providing access to both event times and censoring times. For each original dataset, we generate synthetic versions with controlled censoring rates $\rho \in \{10, 30, 50, 70, 90\}\%$ under three censoring mechanisms: administrative censoring (AC), independent censoring (IC), and covariate-dependent censoring (CDC). For each combination of dataset, censoring mechanism, and censoring rate, we generate 5 independent replications to account for stochastic variability.

All procedures start from the subset of individuals who experienced the event in the original dataset ($\delta_i = 1$), ensuring that the true event times $T_i$ are known for all subjects.

### C.1. Covariate-Dependent Censoring (CDC)

To generate covariate-dependent censoring, we first model the empirical censoring process observed in the original dataset. Specifically, censoring is treated as the event of interest by inverting the indicator, $\delta_i^{(C)} = 1 - \delta_i$, and we fit a Cox proportional hazards model using covariates $x_i$. This yields an estimate of the *conditional censoring survival function*

$$\hat{G}(t \mid x_i) = \widehat{\Pr}(C_i > t \mid X_i = x_i) = \exp\left(-\hat{H}_0(t) \exp(x_i^\top \hat{\beta}_C)\right),$$

where $\hat{H}_0(t)$ is the baseline cumulative hazard and $\hat{\beta}_C$ the fitted regression coefficients. Equivalently, the corresponding censoring CDF is

$$\hat{F}_{C,i}(t) = 1 - \hat{G}(t \mid x_i).$$

**Selection of subjects to censor.** We start from individuals with observed events in the original data ($\delta_i = 1$), so that their true event times $T_i$ are known. For each such individual, we compute the estimated probability of being censored before the event time:

$$\hat{P}_{C,i} = \hat{F}_{C,i}(T_i) = 1 - \hat{G}(T_i \mid x_i).$$

To achieve a target censoring rate $\rho$, we select a subset of individuals to censor using weighted sampling without replacement, with weights proportional to $\hat{P}_{C,i}$.

**Generation of censoring times.** For each selected individual, we sample a censoring time from the individual censoring distribution truncated to $[0, T_i)$. Define the truncated CDF

$$F_i^*(t) = \begin{cases} 0, & t < 0, \\ \dfrac{\hat{F}_{C,i}(t)}{\hat{F}_{C,i}(T_i)}, & 0 \leq t < T_i, \\ 1, & t \geq T_i. \end{cases}$$

We then draw $u_i \sim \mathcal{U}(0,1)$ and set

$$C_i^* = (F_i^*)^{-1}(u_i) = \hat{F}_{C,i}^{-1}(u_i \, \hat{F}_{C,i}(T_i)),$$

which guarantees $C_i^* < T_i$ almost surely. For non-selected individuals, no censoring is applied.

Finally, the observed data are defined as $y_i = \min(T_i, C_i^*)$ and $\delta_i = \mathbf{I}\{T_i \leq C_i^*\}$.

## C.2. Independent Censoring (IC)

Independent censoring is generated using the same procedure as in the covariate-dependent case, except that the censoring distribution is estimated *marginally*, without covariates. Specifically, the censoring survival function $\hat{G}(t) = \widehat{\Pr}(C_i > t)$ is obtained using a Kaplan–Meier estimator fitted on the original dataset (treating censoring as the event of interest), and we define the corresponding CDF $\hat{F}_C(t) = 1 - \hat{G}(t)$. All subsequent steps—weighted sampling to reach the target censoring rate and assignment of censoring times—are identical to Section C.1.

For each selected individual with known event time $T_i$, we sample a censoring time from the marginal censoring distribution truncated to $[0, T_i)$ via inverse transform:

$$C_i^* = \hat{F}_C^{-1}(u_i \, \hat{F}_C(T_i)), \qquad u_i \sim \mathcal{U}(0,1),$$

which guarantees $C_i^* < T_i$ almost surely. This construction yields censoring times that are independent of the covariates by design.

## C.3. Administrative Censoring (AC)

Administrative censoring is generated by simulating staggered study entry with a fixed study end time. For each dataset, we first determine a study close time $S$, initialized as a high quantile (95%) of the true event-time distribution. To reach a target censoring rate $\rho$, we introduce a parameter $\beta \in (0, 1)$ controlling the spread of entry times.

For each individual, an entry time is drawn as

$$E_i \sim \mathcal{U}(0, \beta S),$$

and the administrative censoring time is defined as

$$C_i^* = S - E_i.$$

Observed times and event indicators are then given by $y_i = \min(T_i, C_i^*)$ and $\delta_i = \mathbf{I}\{T_i \leq C_i^*\}$.

The parameter $\beta$ is selected via a bisection search to match the target censoring rate $\rho$ within a specified tolerance. If the target rate cannot be achieved for the initial $S$, the study close time is iteratively reduced until a feasible solution is found. For each target censoring rate, this procedure is repeated independently five times, yielding five replicated datasets with strictly administrative censoring.

## C.4. Event and Censoring Time Distributions

Figure 4 reports empirical cumulative distribution functions (ECDFs) of time variables under ADMINISTRATIVE CENSORING. An ECDF represents, for any time $t$, the proportion of observations with value less than or equal to $t$, and provides a nonparametric summary of the time distribution.

Under AC, censoring predominantly occurs late in follow-up, as it is induced by a fixed study close with staggered entry. Consequently, only a small fraction of individuals are censored before early evaluation horizons. This explains why early-horizon calibration metrics (e.g., $t_{25}$) are less affected under AC: censored individuals have similar contributions under standard (ST) and oracle (OR) evaluations, limiting artificial uniformization effects in calibration bins.

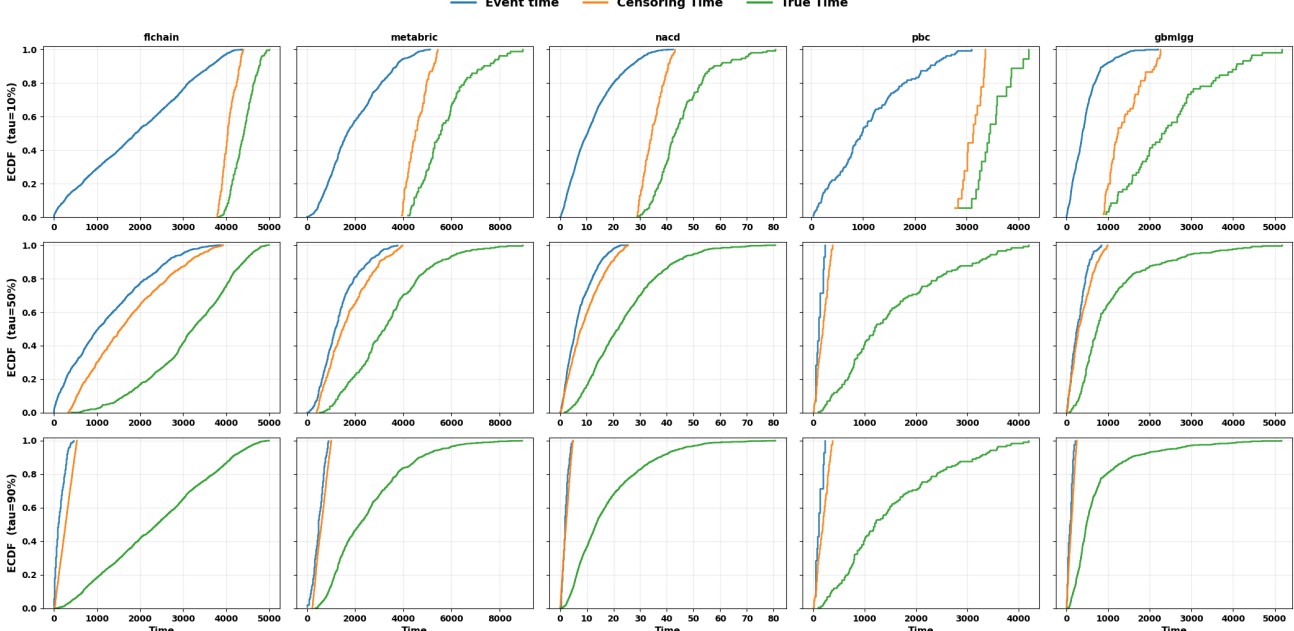

*Figure 4.* **ECDFs of time variables under administrative censoring.** The figure shows empirical cumulative distribution functions of observed event times, observed censoring times, and true event times for censored individuals. Observed event times correspond to individuals with $\delta = 1$, observed censoring times to individuals with $\delta = 0$, and true event times denote the underlying event times of censored individuals, which are available by construction in the semi-synthetic data (Appendix C.3).

# D. Model Implementation and Training Details

We trained seven survival models—Cox Proportional Hazards (COXPH), Weibull Accelerated Failure Time (WEIBULL-AFT), Random Survival Forests (RSF), Neural Multi-Task Logistic Regression (NMTLR), DeepHit, DeepSurv, and Deep Cox Mixtures—on each semi-synthetic dataset generated in Section C. Models were implemented using standard open-source libraries: `lifelines` (COXPH, WEIBULL-AFT), `scikit-survival` (RSF), `pycox` and `torchtuples` (DEEPHIT, NMTLR, DEEPSURV), and `auton-survival` (DEEP COX MIXTURES). All experiments were run in Python 3.10.

## D.1. Preprocessing and Time Discretization

Categorical covariates (when present) were one-hot encoded. Continuous covariates were normalized within each outer fold, with the scaler fitted on the training split and applied to the corresponding test split.[6]

For discrete-time neural models (DEEPHIT and NMTLR), event times were discretized into $M = 100$ bins defined from

---

[6]Continuous covariates are scaled within each outer fold: the scaler is fitted on the training split and applied to the corresponding test split.

the training split using quantile-based cut points (i.e., approximately equal probability mass). This discretization yields a discrete-time event distribution $\hat{f}(t_k \mid x)$ over the bins, from which we recover a survival function by cumulative product,

$$\hat{S}(t_j \mid x) = \prod_{k \leq j} \big(1 - \hat{f}(t_k \mid x)\big).$$

Predicted survival curves are evaluated on the resulting cut grid and linearly interpolated when metrics require values at intermediate times. For DEEPSURV and DEEP COX MIXTURES, survival curves were obtained from the fitted continuous-time models and evaluated on the corresponding prediction time grids before computing the same evaluation metrics.

### D.2. Training Protocol and Hyperparameter Selection

For DEEPHIT, NMTLR, DEEPSURV, DEEP COX MIXTURES, and RSF, we used nested cross-validation to avoid optimistic bias due to hyperparameter tuning: a 5-fold outer loop for evaluation and a 3-fold inner loop for model selection. Hyperparameters were optimized using Optuna with a TPE sampler, targeting predictive discrimination on validation folds. In particular, for neural models we optimized the HARRELL'S concordance criterion computed from a cumulative-risk score derived from predicted survival curves.

For the tuned models, we optimized a set of model-specific hyperparameters reflecting their respective inductive biases (Table 2). For DEEPHIT, the search space included architectural regularization via dropout, the loss-specific parameters $(\alpha, \sigma)$ controlling the trade-off between likelihood and ranking components, as well as the learning rate and batch size. NMTLR followed a similar neural setup, with tuning limited to dropout, learning rate, and batch size, reflecting its simpler loss formulation. For DEEPSURV, we tuned dropout, learning rate, weight decay, and batch size. For DEEP COX MIXTURES, we tuned the number of mixture components, hidden-layer architecture, loss parameter $\gamma$, smoothing factor, activation usage, learning rate, batch size, and the number of optimization iterations. For RSF, we tuned standard tree-based hyperparameters, including the number of trees, maximum depth, minimum samples per split and leaf, and the number of features considered at each split.

In contrast, COXPH and WEIBULL-AFT were fitted on each outer training split without hyperparameter optimization (beyond default regularization where applicable) and evaluated on the corresponding outer test split.

### D.3. Replications and Evaluation Modes

For each dataset, censoring mechanism, and target censoring rate, we generated 5 independent data replications (Section C). For each replication, we ran the full training and evaluation pipeline described above. In each outer fold, after selecting hyperparameters from the inner loop, the model was retrained on the full outer training data and evaluated on the held-out outer test fold.

Each evaluation was performed under two protocols: *standard* evaluation uses the observed right-censored test outcomes $(y_i, \delta_i)$, whereas *oracle* evaluation replaces all test times by the fully observed event times $T_i$ (with event indicators set to 1). This yields paired metric values per fold and replication, enabling controlled quantification of censoring-induced distortions.

### D.4. Prediction Storage and Reproducibility

For each configuration (dataset, mechanism, rate, replication, outer fold), we store predicted survival functions on the model's prediction grid, along with fold indices and the corresponding hyperparameters selected by Optuna. Metric computation is performed from these saved survival curves using the evaluation code described in Appendix A, ensuring that all reported results can be reproduced from stored predictions and configuration files.

## E. Additional Analysis

### E.1. Ipcw Variants of Concordance Metrics

This appendix reports additional results for IPCW-corrected variants of the C-index. For both Harrell's and Antolini's C-index, we consider IPCW corrections based on Kaplan–Meier and Cox models for the censoring distribution; for Harrell's C-index, the Kaplan–Meier–based variant corresponds to the Uno C-index.

These variants are included to assess whether inverse probability weighting mitigates the censoring-induced distortions

*Table 2.* Hyperparameter search spaces for the tuned models.

| MODEL | HYPERPARAMETER | SEARCH SPACE |
|---|---|---|
| DEEPHIT | DROPOUT | $[0, 0.5]$ |
| | LOSS PARAMETER $\alpha$ | $[0.1, 1.0]$ (STEP $0.1$) |
| | LOSS PARAMETER $\sigma$ | $[0.01, 0.2]$ (STEP $0.01$) |
| | LEARNING RATE | $\{10^{-1}, 10^{-2}, 10^{-3}\}$ |
| | BATCH SIZE | $\{32, 64, 128\}$ |
| NMTLR | DROPOUT | $[0, 0.5]$ |
| | LEARNING RATE | $\{10^{-1}, 10^{-2}, 10^{-3}\}$ |
| | BATCH SIZE | $\{32, 64, 128\}$ |
| DEEPSURV | DROPOUT | $[0, 0.5]$ |
| | LEARNING RATE | $\{10^{-3}, 10^{-4}\}$ |
| | WEIGHT DECAY | $[10^{-6}, 10^{-2}]$ (LOG SCALE) |
| | BATCH SIZE | $\{32, 64, 128\}$ |
| DEEP COX MIXTURES | NUMBER OF MIXTURE COMPONENTS | $\{2, 3, 4\}$ |
| | HIDDEN LAYERS | $\{[32], [64], [64, 32]\}$ |
| | LOSS PARAMETER $\gamma$ | $\{1, 10, 100\}$ |
| | SMOOTHING FACTOR | $\{10^{-4}, 10^{-3}, 10^{-2}\}$ |
| | ACTIVATION USAGE | $\{$FALSE, TRUE$\}$ |
| | LEARNING RATE | $\{10^{-3}, 10^{-4}\}$ |
| | BATCH SIZE | $\{32, 64, 128\}$ |
| | NUMBER OF ITERATIONS | $\{50, 100, 200\}$ |
| | OPTIMIZER | $\{$ADAM$\}$ |
| RSF | NUMBER OF TREES | $[100, 400]$ (STEP $50$) |
| | MAXIMUM DEPTH | $[3, 16]$ |
| | MINIMUM SAMPLES PER SPLIT | $[2, 20]$ |
| | MINIMUM SAMPLES PER LEAF | $[1, 10]$ |
| | MAX FEATURES | $\{$SQRT, LOG2$\}$ |

observed for global concordance measures.

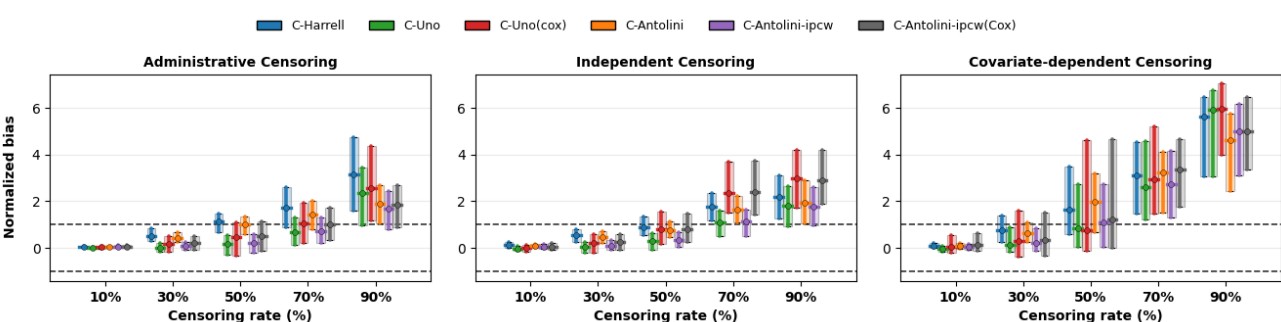

*Figure 5.* Normalized bias of IPCW-based concordance metrics as a function of the censoring rate under (AC), (IC), (CDC). Results are shown for C-Uno and IPCW variants of C-Antolini using Kaplan–Meier and Cox models for the censoring distribution.

Across all mechanisms, Figure 5 shows that IPCW-concordance based variants exhibit bias patterns that remain closely aligned with those of their non-IPCW counterparts, particularly at moderate to high censoring levels. Moreover, Cox-based IPCW variants display systematically larger variability, independently of the censoring mechanism and censoring rate, than their KM-based counterparts, indicating reduced numerical stability of inverse probability weights across datasets and replications.

Under non-informative censoring (AC and IC), using a Kaplan–Meier estimator for the censoring distribution yields a modest but consistent reduction in bias compared to Cox-based IPCW, for both Harrell's and Antolini's concordance. Nevertheless, even in this favorable setting, IPCW-based concordance measures remain noticeably biased at high censoring rates ($\rho \geq 70\%$), indicating that inverse weighting alone does not fully correct distortions in global ranking performance.

Under covariate-dependent censoring (CDC), where Cox-based IPCW is correctly specified in principle, we do not observe a meaningful bias reduction compared to KM-based IPCW. Both implementations exhibit comparable levels of bias, suggesting that accurate modeling of the censoring distribution is insufficient to restore reliable concordance when censoring depends on covariates that are also predictive of the event.

Overall, these results indicate that IPCW corrections—regardless of whether they are KM- or Cox-based—do not fundamentally alter the behavior of global concordance measures under strong censoring. While careful specification of the censoring model can slightly attenuate bias under non-informative censoring, IPCW-based C-indices remain substantially more sensitive to censoring than fixed-horizon concordance measures $C(t_q)$, whose robustness is analyzed in the main paper.

### E.2. Cox-Based Ipcw IBS Variants

We report additional results for *Cox-based IPCW* variants (Appendix A.2) of the Integrated Brier Score (IBS), omitted from the main paper for conciseness. These variants differ from the KM-based IPCW IBS shown in the main text only in the estimation of the censoring distribution $G(t)$, which is here obtained from a Cox proportional hazards model rather than a Kaplan–Meier estimator. Figure 6 shows the corresponding normalized-bias patterns under the three censoring mechanisms.

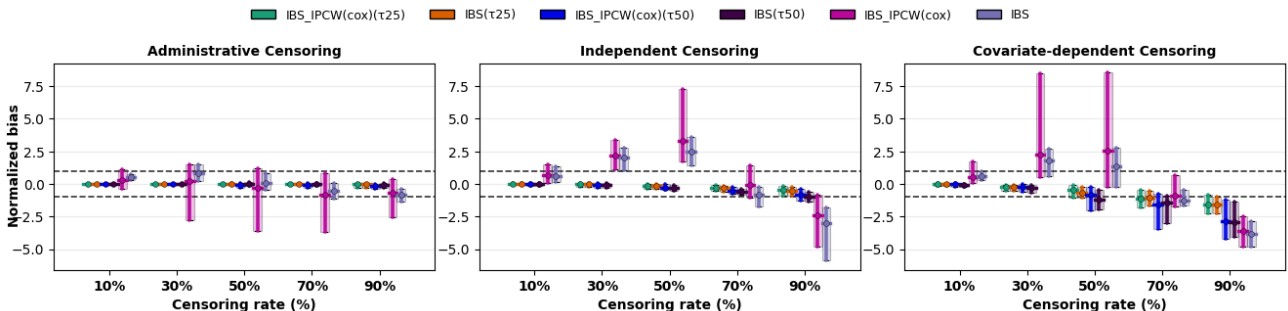

*Figure 6.* Normalized bias of Cox-based IPCW IBS variants across censoring mechanisms and censoring rates. We report the full-horizon IPCW IBS and its truncated-horizon variants at fixed horizons ($t_{25}$, $t_{50}$.)

Under AC, Cox-based IPCW IBS variants do not exhibit a systematic reduction of bias compared to their KM-based counterparts reported in the main paper. While average bias remains small across censoring rates and truncated variants ($t_{25}$, $t_{50}$) are the most stable in expectation, Cox-based estimates display substantially higher variability. This increased variance offsets any potential benefit from modeling covariate effects in the censoring distribution. Consequently, under non-informative administrative censoring, the Kaplan–Meier–based IPCW IBS provides a more stable and practically preferable alternative.

A similar conclusion holds under IC. Cox-based IPCW IBS broadly follows the same average bias trends as the KM-based variants reported in the main text, with truncated horizons remaining the most robust in expectation. However, estimating the censoring distribution via a Cox model leads to substantially higher variability, particularly at high censoring rates. For the full-horizon IPCW IBS, this increased variance is accompanied by slightly larger bias under heavy censoring, reflecting the instability of inverse-probability weights when the censoring model is more complex. As a result, although qualitative bias patterns are similar, KM-based IPCW IBS offers a markedly more stable and practically preferable choice under independent censoring.

Under CDC, where Cox-based IPCW is theoretically better aligned with the censoring mechanism, we do not observe a systematic reduction of bias. For truncated horizons, Cox-based IPCW IBS behaves essentially identically to the KM-based variants shown in the main paper. For the full-horizon IBS, bias is slightly more pronounced, indicating that weight explosion is even stronger when estimating the censoring distribution via Cox models. This mirrors the behavior observed for IPCW-based concordance metrics and suggests that, under informative censoring, improved censoring-model specification alone is insufficient to mitigate IBS bias.

Overall, Cox-based IPCW IBS variants do not yield systematic bias reductions relative to their KM-based counterparts. While truncated-horizon estimates follow similar average trends, Cox-based weights introduce substantially higher variability, and full-horizon IBS is particularly sensitive to weight instability. As a result, KM-based IPCW IBS provides a more stable

and practically reliable choice across censoring mechanisms.

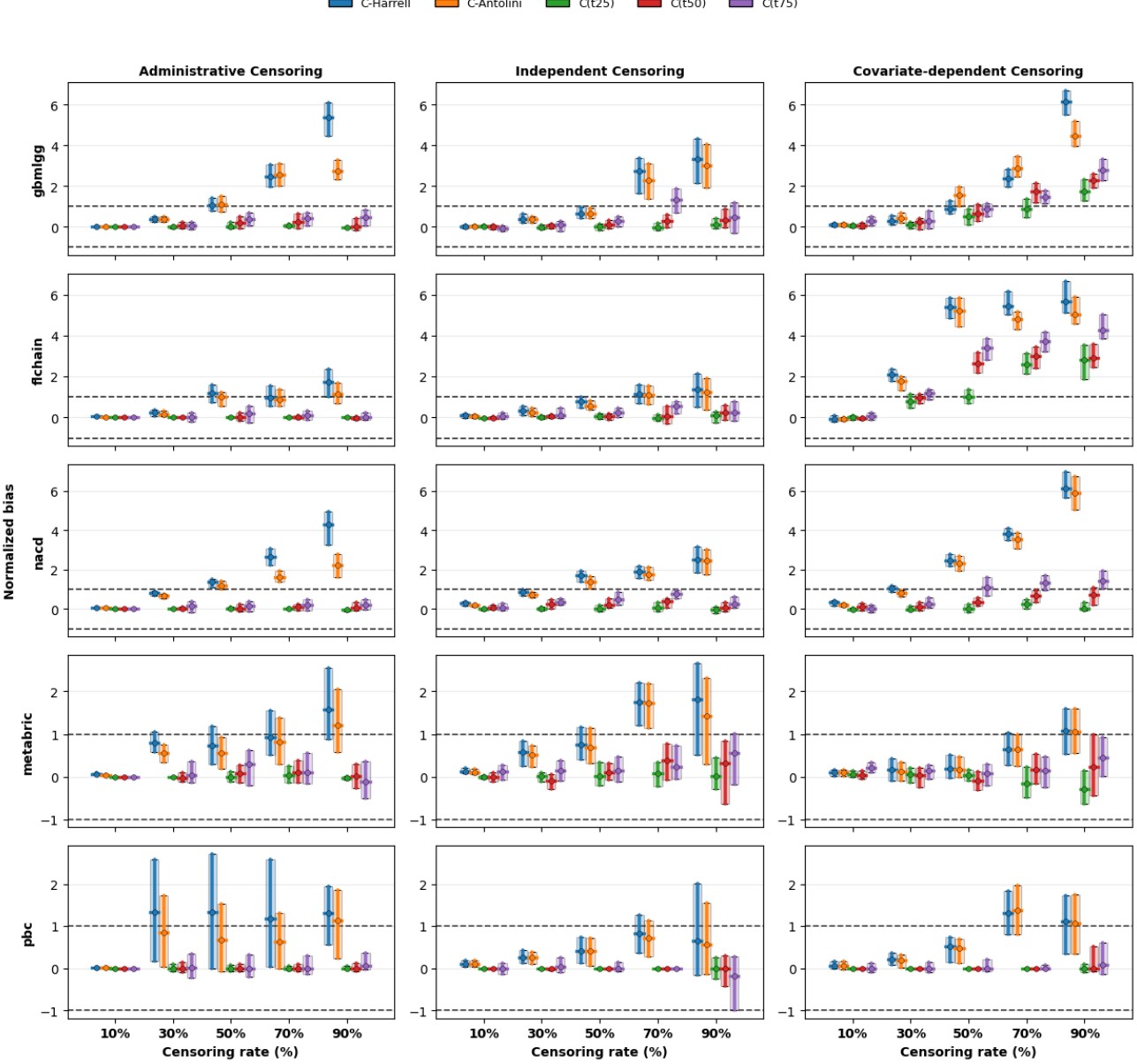

*Figure 7.* Normalized bias under controlled censoring, reported **per dataset** for the **concordance-family** metrics.

### E.3. Additional Results Across Datasets

#### E.3.1. METRIC BIAS UNDER CONTROLLED CENSORING.

This appendix complements the aggregated results reported in the main paper by showing the corresponding normalized-bias illustrations separately for each dataset. The per-dataset results are reported for the three metric families: concordance-family metrics in Figure 7, IBS-family metrics in Figure 8, and calibration-family metrics in Figure 9.

#### E.3.2. PRESERVATION OF MODEL RANKING UNDER CONTROLLED CENSORING

Figure 10 extends the analysis presented in Figure 2 of the main paper to *all* evaluation metrics considered in this study. For each metric, we report both Top-1 consistency (odd rows) and the dominance-based global ranking agreement (even rows),

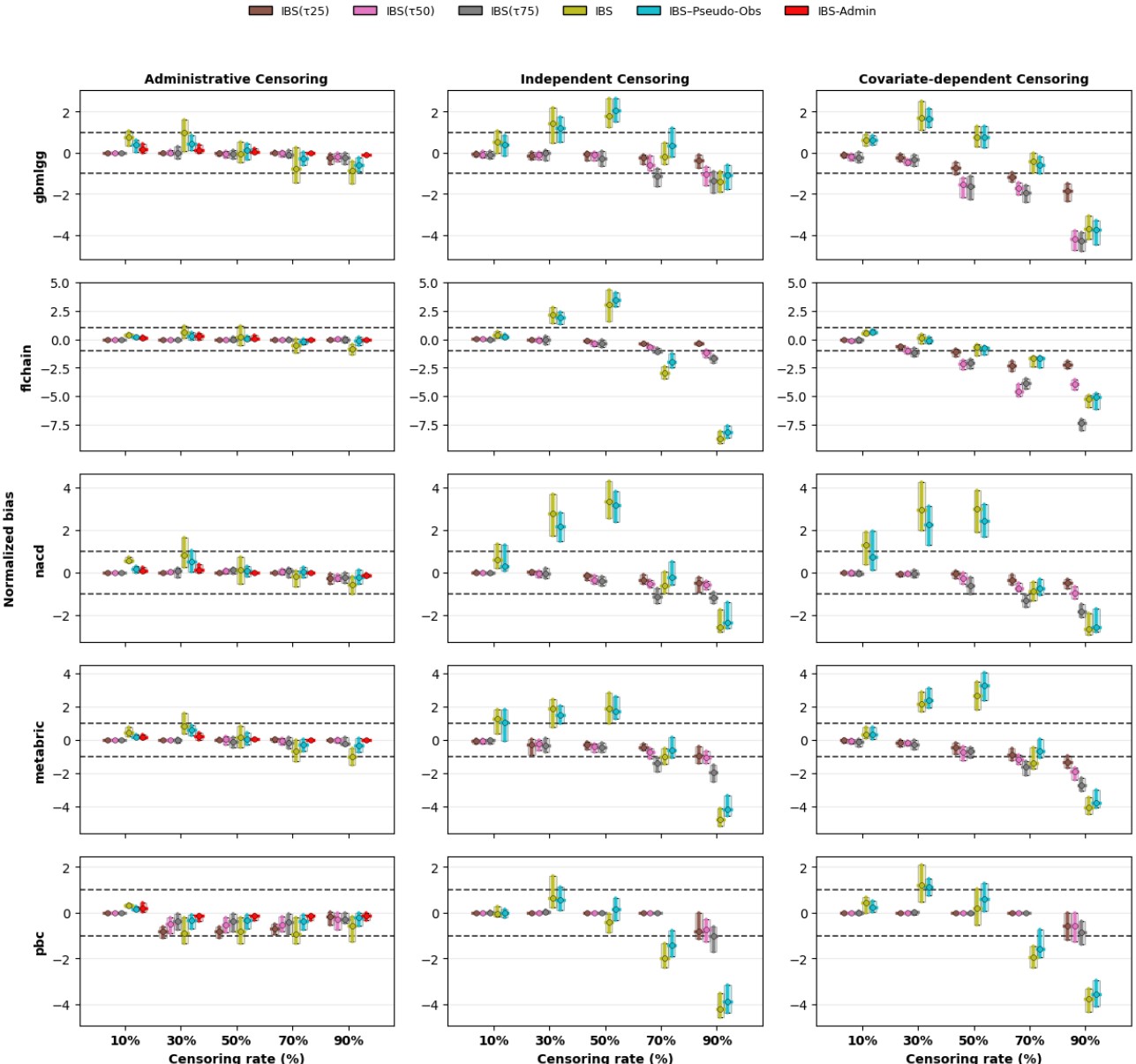

*Figure 8.* Normalized bias under controlled censoring, reported **per dataset** for the **IBS-family** metrics.

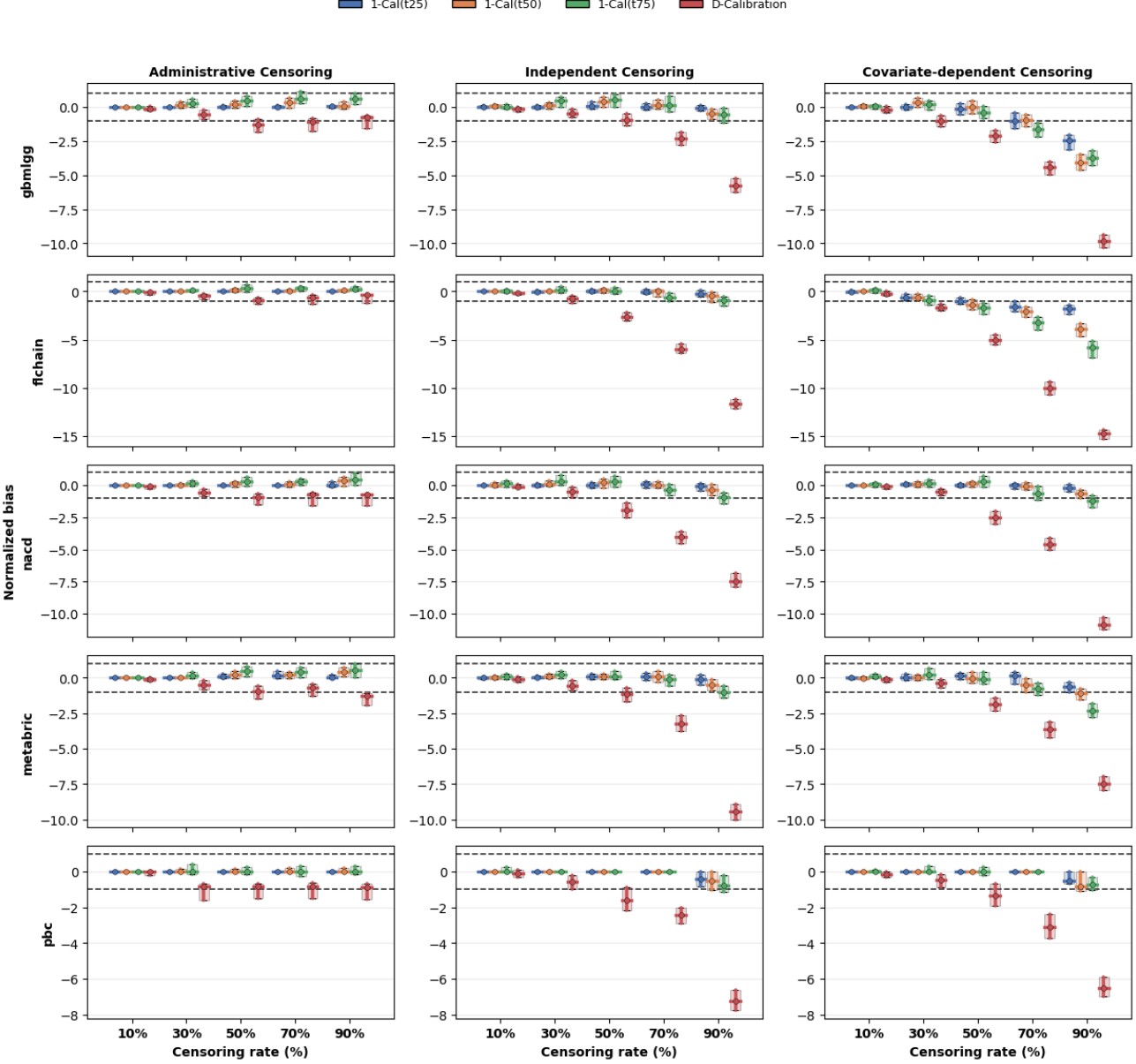

*Figure 9.* Normalized bias under controlled censoring, reported **per dataset** for the **calibration-family** metrics.

computed on statistically comparable model pairs, across censoring rates and censoring mechanisms.

These supplementary results confirm that conclusions drawn from the main figure are not specific to a particular metric family: metrics can exhibit low numerical bias while still yielding unstable Top-1 selections, whereas dominance-based agreement often remains high when the global ordering is preserved. The appendix figures therefore provide a complete view of ranking stability across discrimination, probabilistic accuracy, and calibration criteria.

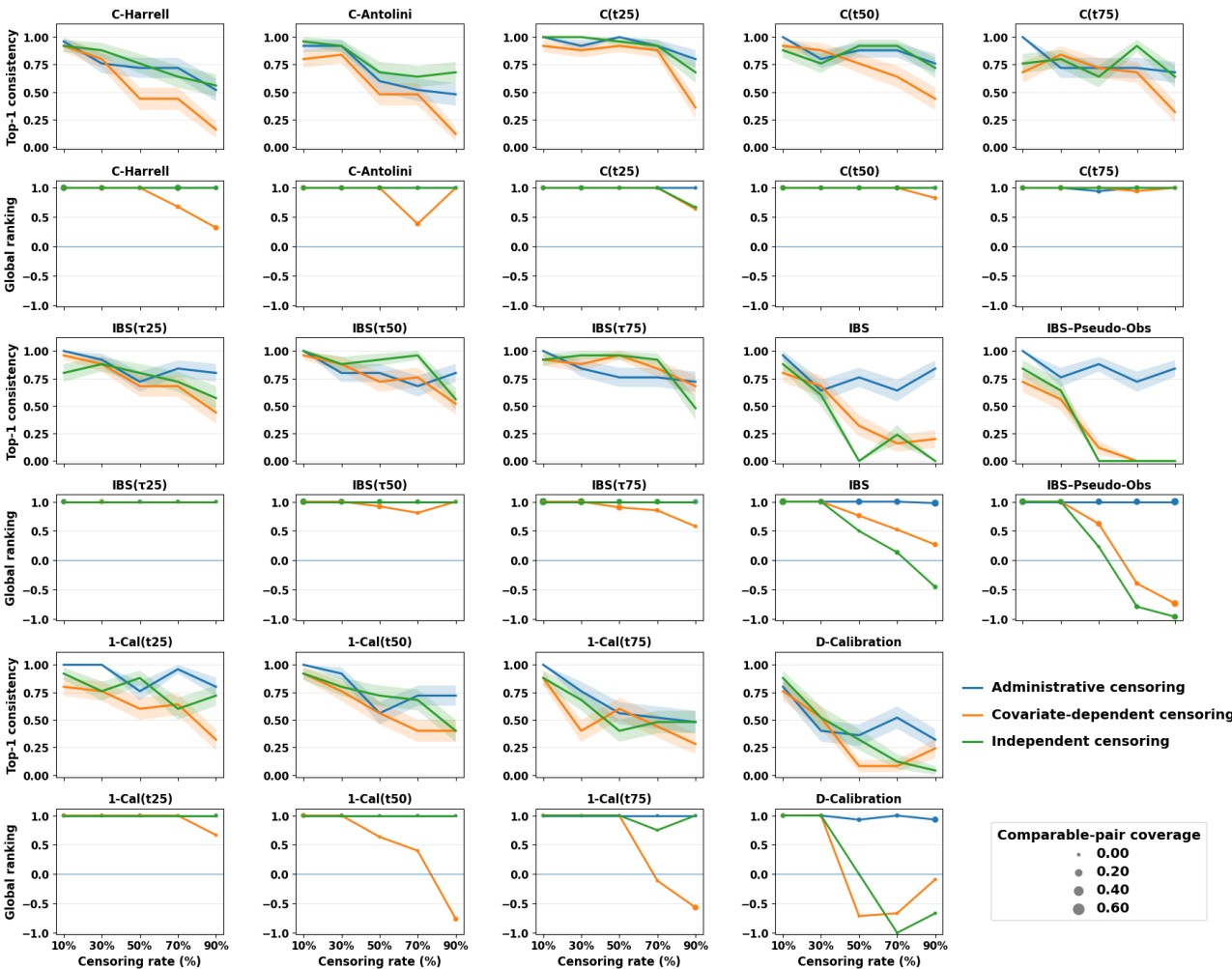

*Figure 10.* **Preservation of model ranking for all evaluation metrics.** For each metric, Top-1 consistency (odd rows) and dominance-based global ranking agreement (even rows) are shown as a function of the censoring rate under administrative, independent, and covariate-dependent censoring. Marker size reflects comparable-pair coverage.

