# OpenReview forum: "When Can We Trust Survival Model Evaluation ?"
_ICML.cc/2026/Conference — ICML 2026 regular_

### Official Review · Reviewer_YRCs · 2026-03-01

**Soundness:** 3
**Presentation:** 3
**Significance:** 3
**Originality:** 3
**Overall Recommendation:** 5
**Confidence:** 3

**Summary:**

This manuscript focuses on the concept of reliability of survival model evaluation under varying censoring mechanisms and rates. The authors use a semi-synthetic framework to compare a censored evaluation and an oracle reference. The core contribution is a large-scale experimental study with multiple survival datasets, censoring strategies and evaluation metrics. The two primary evaluation axes are normalized metric bias and ranking preservation. Practical suggestions are given to future studies based on those analyses.

**Compliance With Llm Reviewing Policy:**

Affirmed.

**Final Justification:**

I will keep my score

**Key Questions For Authors:**

1. Would there be a change in conclusion if models were not tuned for C-Harrell but for IBS or calibration?
2. Whether the conclusions remain the same when the number of features used for model prediction change? Does a small model get more influenced than a larger model by censoring rate and mechanism?
3. To what extent do models learn from censoring information instead of the real pathological signals? Is there a model overfitting problem on censoring patterns?
4. Would it be possible to translate the empirical findings into more statistically concrete guidance for developing, reporting, and interpreting survival models? The current conclusion provides some insight but may be too high-level. For example, can the authors propose quantitative thresholds (e.g., minimum comparable-pair coverage, censoring rate ranges, IPCW weight variability diagnostics) beyond which certain metrics should be considered unreliable? Are there principled criteria for choosing truncation horizons based on the censoring distribution?

**Limitations:**

While the paper includes a discussion of methodological limitations, it does not address potential negative societal impacts.

**Strengths And Weaknesses:**

Strengths:
1. Clear experimental setup. The authors did a great job introducing the dataset, evaluation metrics, and censoring control. The separation of censoring rate and mechanism contributes to clear demonstration of metric computation distortion.
2. This paper carefully distinguishes between bias and ranking stability. Through well-designed experiments, they conclude that small numerical bias can still lead to unstable Top-1 selection and provide practical suggestions for future research.
3. They covered 5 datasets, 3 censoring mechanisms, 5 censoring rates, 5 models and different evaluation metrics. This broad coverage strengthens their claims about censoring issues.

Weaknesses:
1. Semi-synthetic data may not reflect the real complexity. The conclusion may not be generalizable to real-world data where there are unobserved confounding factors.
2. The paper is based on empirical evidence from generated datasets. A stronger conclusion would be from more theoretical analyses explaining, e.g. why there are certain changes in bias and why global ranking may remain stable but Top-1 fails.

---

> ### Author Rebuttal · Authors · 2026-03-30
>
> We thank the reviewer for the positive assessment and for highlighting the clarity of the setup and the distinction between numerical bias and ranking stability.
>
> **Weakness 1**
> We fully agree that our semi-synthetic framework abstracts away from some real-world complexities, such as unobserved confounding, competing risks, or time-varying covariates. This is a deliberate trade-off: the controlled setup is precisely what makes the oracle comparison possible, and therefore allows us to isolate censoring-induced distortion in metric computation. A purely real-world study would not provide access to true event times and would thus preclude this analysis. We will strengthen this limitation in the paper and explicitly frame our conclusions as necessary conditions for reliable evaluation, not sufficient ones. We will also mention as a future direction a partially hidden censoring setting, where censoring depends on variables used to generate censoring but omitted from model training, to better mimic unobserved censoring drivers.
>
> **Weakness 2**
> We agree that stronger theory would further strengthen the paper. However, deriving general bias results for all families of survival metrics under realistic censoring—especially covariate-dependent censoring—is non-trivial and remains largely open. Our contribution is therefore primarily to provide a large-scale controlled empirical characterization that can serve as a basis for future theory. For some patterns, we do provide mechanism-level explanations. For instance, the IBS sign change arises because at low censoring ST is slightly pessimistic, whereas at high censoring the effective estimable horizon contracts and the metric becomes dominated by earlier times, where prediction errors are typically smaller, yielding overly optimistic values.
>
> **Q1**
> We did examine this point before submission through preliminary but incomplete checks using alternative tuning objectives, including IBS-type and calibration-oriented criteria. Although these experiments were not complete enough for systematic inclusion, they pointed to the same main qualitative result: the central patterns reported in the paper remained unchanged when the optimization metric was varied. In particular, for a fixed set of trained models, the ST/OR comparison still revealed censoring-induced distortion at the evaluation level. Different tuning objectives may change absolute scores and sometimes the oracle ranking itself, but they did not alter the main conclusions of the paper. Since our goal is to isolate censoring effects on metric computation rather than compare metric-specific training objectives, we kept a single training protocol in the main paper for clarity and comparability. We agree, however, that this is worth mentioning more explicitly in the discussion.
>
> **Q2**
> We did not vary feature dimensionality as an independent factor, but our experiments already span datasets with 13 to 79 features and model classes ranging from parametric (CoxPH, Weibull) to non-parametric (RSF) and neural models (DeepHit, NMTLR). The qualitative trends are consistent across this range, suggesting that the main driver is the censoring structure rather than model capacity alone. We will state this more clearly and mention systematic sparse-vs-high-capacity comparisons as future work.
>
> **Q3**
> Yes, this is a real risk under covariate-dependent censoring. In that setting, models may partly exploit censoring-related structure rather than only event-related signal. Our paper documents the **consequence** of this at evaluation time: standard metrics can then reward models in ways that are misaligned with oracle event-time performance. Fully disentangling model-learning effects from metric-computation effects would require additional cross-design experiments (e.g., training under one censoring regime and evaluating under another), which we view a follow-up direction.
>
> **Q4**
> Yes. We agree the recommendations can be made more operational. In the revision, we will add more concrete diagnostics, while keeping them explicitly empirical rather than universal:
> (i) treat large normalized bias (e.g., >1 IQR) as a warning sign; in our results this typically appears from high censoring under AC/IC and earlier under CDC;
> (ii) always report comparable-pair coverage for concordance metrics;
> (iii) monitor IPCW weight instability for IPCW-based metrics;
> (iv) prefer truncation horizons supported by sufficient event information and stable censoring estimates, rather than using full horizons by default.
>
> Finally, we will add a short societal-impact note: unreliable survival evaluation under censoring can lead to overconfident model selection in high-stakes settings such as healthcare and reliability.

---

> > ### Author Rebuttal · Reviewer_YRCs · 2026-04-01
> >
> > I would like to thank authors for the explanation. I will keep my score.

---

### Official Review · Reviewer_LEEJ · 2026-03-11

**Soundness:** 4
**Presentation:** 3
**Significance:** 3
**Originality:** 2
**Overall Recommendation:** 4
**Confidence:** 4

**Summary:**

The paper studies the impact of different kinds of censoring (administrative, independent, covariate-dependent) and rates of censoring impact standard evaluation practices. For this, the authors use a set of established survival modeling datasets, filter them for individuals with recorded events, and then construct semi-synthetic datasets by reintroducing synthetic censoring for this subset of patients. They analyze the impact of the kind and rate of censoring, particularly looking at numerical bias and preservation of ranking in evaluation metrics. Importantly, the authors show that top-1 model selection can be very fragile, yet global rankings (taking statistical significance of performance comparisons into account) are often preserved.

**Compliance With Llm Reviewing Policy:**

Affirmed.

**Final Justification:**

The paper is technically solid, well executed, and clearly written, with a useful controlled study of how censoring rate and mechanism affect survival-model evaluation. The rebuttal addressed several of my secondary concerns, especially by clarifying the recommendations around truncated metrics and discussing the semi-synthetic design more carefully. However, my concern about the Top-1 framing remains only partially resolved: the new analyses suggest that part of the reported instability reflects cases where there is no statistically reliable best model in the first place, which weakens the headline claim even though it supports the broader message that model selection becomes less reliable under heavier censoring.

**Key Questions For Authors:**

* In the construction of the semi-synthetic datasets, you first filter to only subjects with an event, then add synthetic censoring. Can you please comment on and discuss in the paper to what extent this may introduce systematic biases and whether this affects your evaluation in any way?
* Given that you cite Lillelund et al. (2025a) extensively and their copula-based metrics are specifically designed for the dependent censoring setting where your results show the largest failures, why were these metrics not included in your evaluation?

**Limitations:**

yes

**Strengths And Weaknesses:**

## Strengths

* The paper is well motivated and well written.
* Comprehensive set of evaluation metrics.
* The proposed measures are well designed, in particular the normalized bias measure is intuitive, robust, and well suited for comparison across heterogeneous datasets.
* Likewise, the global agreement score that discards non-significant pairs is very well designed.
* The study uses a proper nested cross-validation setup with hyperparameter optimization, which is important for deriving robust insights from the used (rather small) survival modeling datasets.
* Given how noisy performance evaluations on the used datasets tend to be, it is reassuring how consistent the reported results are across the datasets.

## Weaknesses

* To me, the main takeaway result appears to be the instability of the top-1 model selection, as often the model with the best performance is chosen for further use or downstream experiments. But the top-1 consistency metric treats any swap as a failure, without taking statistical significance of the differences into account. Conversely, the global agreement analysis shows that when taking statistical significance into account, results are much more consistent. A "statistical Top-1" that checks whether the oracle-best and standard-best are significantly different would be more informative and could substantially change the headline finding.
* The practical recommendations i and ii are sound but arguably well-known best practices, and are not particularly novel findings. Recommendation (iii) "prefer early-horizon or truncated metrics" deserves more scrutiny: truncation reduces bias but also reduces comparable pairs and may increase variance. Whether the bias-variance tradeoff actually favors truncation for model selection is not systematically analyzed. Additionally, restricting to early events may not capture clinically relevant time horizons depending on the application, which could introduce other problematic biases.

### Minor points

* Figure 1 caption: The dashed reference line (presumably at normalized bias = 1) is not explained in the caption or legend.

---

> ### Author Rebuttal · Authors · 2026-03-30
>
> We thank the reviewer for the careful reading and for the positive assessment of the paper’s motivation and methodology.
>
> **Weakness 1**
> Following your suggestion, we implemented a significance-aware statistical Top-1 analysis using the same CV-corrected test as in our response to Reviewer k2fd. For each dataset, censoring mechanism, censoring rate, and replication, we compared the top-1 and top-2 models in both ST and OR, and retained only cases where this gap was significant in both settings.
>
> This sharpens the interpretation of the original Top-1 result, but also reveals an important limitation: in many settings, especially for some calibration metrics and under heavier censoring, no reliable winner can be identified because the top-2 gap is not significant. In our experiments, 95/225 metric × censoring-rate × mechanism configurations had no comparable cases after this filtering. So the issue is not only that Top-1 may change, but also that many apparent best-model differences are simply too weak to support a reliable selection.
>
> Because this first definition becomes sparse, we also considered a second significance-aware Top-1 variant: we identify the oracle-best model and test whether its mean standard score remains within a confidence interval around its average oracle score. This version remains usable in almost all cases (6741/6750 usable cases; mean coverage 0.999) and leads to the same conclusion: Top-1 preservation degrades steadily with censoring, from 0.969 at 10% to 0.483 at 90%, with the strongest degradation under covariate-dependent censoring.
>
> We therefore believe that both analyses support the same overall message: once statistical uncertainty is taken seriously, the issue is not only that Top-1 rankings may change, but also that under censoring the evidence for declaring a unique winner is often weak. We would be happy to report both statistical Top-1 analyses in the appendix.
>
> **Weakness 2.**
> This is a fair critique. Following your suggestion, we ran an additional variance analysis on the standard metric values across across evaluation horizons for horizon-specific C-index, IBS, and 1-Calibration. The tradeoff is clearly metric-family dependent, and for IBS also censoring-rate dependent, while remaining qualitatively similar across mechanisms.
>
> For 1-Calibration, shorter horizons generally reduce bias, and variance is also usually lower at earlier horizons. For horizon-specific C-index, variance instead increases at shorter horizons, consistent with fewer comparable pairs. For IBS, the pattern is less monotonic: at low censoring, variance tends to decrease with the horizon, while at high censoring it increases with the horizon. We therefore agree that recommendation (iii) should be softened: truncated metrics are a pragmatic mitigation strategy when late-horizon evaluation becomes unreliable, but not a universally preferable choice for model selection. We would be happy to add this variance analysis in the appendix.
>
> We also agree that recommendations (i) and (ii) may sound like best practices. Our point is precisely that, while sensible, they are still not consistently followed in current survival benchmarking, where fold-averaged scores are often reported without discussing how censoring affects metric reliability. We therefore think it is useful to document empirically why these practices matter.
>
> **Q1**
> This is a methodological point that we will clarify more explicitly. By construction, our semi-synthetic datasets retain only subjects with observed events, so that true event times are known for all individuals before censoring is reintroduced. This may shift the covariate distribution relative to the full original cohort, especially if event occurrence is strongly covariate-dependent. Two factors mitigate this concern: (i) covariate distributions are preserved within the selected subset, so the feature–outcome relationship is not directly altered; and (ii) the simulated censoring mechanisms, especially CDC, are fitted on the original dataset and therefore reflect empirical censoring patterns. Most importantly, ST and OR are computed on the same generated samples, so the paired comparison remains valid for isolating censoring-induced metric distortion. We will add a short discussion of this selection effect and its implications.
>
> **Q2**
> This is a relevant point, we did not include these copula-based metrics because they require knowledge or modeling of the dependence between event and censoring times, which is precisely the kind of information that is usually not available in practice. Our aim here was to focus on metrics that are both commonly used and practically usable under realistic data constraints. That said, we agree that our protocol could be used to test whether such dependence-aware metrics indeed show lower bias under controlled dependent censoring. We will mention this explicitly in the discussion as a valuable extension

---

> > ### Author Rebuttal · Reviewer_LEEJ · 2026-04-01
> >
> > I thank the authors for the thorough rebuttal. Most of my concerns have been adequately addressed.
> >
> > However, regarding my main point about the Top-1 metric, I feel the new significance-aware analysis actually reinforces rather than resolves my concern. As I noted in my review, the original Top-1 metric treats any swap as a failure without accounting for statistical significance. The new analysis reveals that in many configurations, the top models are simply not significantly separable, which confirms my suspicion that the headline finding conflates two distinct phenomena: censoring changing which model appears best, and there being no reliably best model in the first place. The latter is arguably the more common and less novel finding, and weakens what was for me the key takeaway from the paper.
> >
> > I will retain my current rating of a weak accept.

---

> > > ### Author Response · Authors · 2026-04-04
> > >
> > > Thank you for this helpful clarification, and for maintaining your score. We agree that the significance-aware analyses help distinguish two related phenomena that were not sufficiently separated in our original presentation: (i) censoring can alter which model appears best, and (ii) in many settings, the apparent top models are not statistically distinguishable in the first place.
> > >
> > > Our intent is not to deny this distinction, but to make it explicit and incorporate it into the revised message of the paper. In the first significance-aware analysis, many configurations indeed become non-comparable, especially under heavier censoring. This is an important result in itself, as it shows that censoring does not only affect the identity of the selected top model, but can also weaken the evidence for any unique winner. Importantly, we view both phenomena as consequences of censoring: censoring can change the apparent winner, but it can also reduce the statistical separability between top models. In that sense, the sharp drop in comparable cases under heavier censoring is itself part of the main takeaway.
> > >
> > > At the same time, this is not the whole story. In our second, more stable analysis, designed to remain informative when pairwise top-model comparisons become too sparse, we still observe a clear deterioration with censoring: the preservation rate decreases from 0.969 at 10% censoring to 0.784 at 30%, 0.664 at 50%, 0.586 at 70%, and 0.483 at 90%. The same degradation appears across censoring mechanisms, with mean stability 0.835 under administrative censoring, 0.674 under independent censoring, and 0.584 under covariate-dependent censoring.
> > >
> > > We therefore view the refined analysis as clarifying the interpretation of the original result. More precisely, the issue is not only that apparent Top-1 selections may change under censoring, but also that in many settings the evidence for any uniquely best model becomes too weak to support a reliable choice. From this perspective, the broader practical implication is that model-selection conclusions become less reliable as censoring increases. We agree that this distinction should be made much clearer in the paper, and we would revise the discussion accordingly.

---

### Official Review · Reviewer_k2fd · 2026-03-12

**Soundness:** 2
**Presentation:** 3
**Significance:** 3
**Originality:** 3
**Overall Recommendation:** 4
**Confidence:** 4

**Summary:**

The authors study the impact of censoring on different evaluation metrics for survival models. They vary both the censoring rate as well as the censoring mechanism, and observe the impact of these changes on a variety of different measures, including 3 broad categories: concordance-based metrics such as Harrell's C-index; integrated Brier score-type metrics; and calibration metrics. They compare the metrics as evaluated on censored vs. oracle uncensored data, examining the bias in the metric values themselves (i.e., do models appear more or less favorable when they are evaluated on censored data), as well as the stability of the model *rankings* induced by the metrics (i.e., do the metrics give a consistent "winner" among a set of candidate models as censoring increases). Based on their findings, they give some practical recommendations for evaluation of survival models in practice.

**Compliance With Llm Reviewing Policy:**

Affirmed.

**Final Justification:**

I believe that Weaknesses 2 and 3 have been fully resolved, and the intuition provided for the Key Question is reasonable and the discussion is appreciated. Using the CV-corrected t-test was especially important, and I am glad to see the candid discussion of what changed (about half as many significant model pairs) but that the overall qualitative results are the same. Based on these improvements, I have increased my score. I will refrain from increasing it further because while I believe that the arguments provided re: Weaknesses 1 and 4 have some merit, I still believe adressing these concerns in full would be necessary for showing that the results of the paper are far-reaching/significant in scope.

**Key Questions For Authors:**

Do the authors have some intuition for why the *direction* of the bias for the IBS metrics changes as censoring goes from low to medium to high? This seems like a very unexpected phenomenon.

**Limitations:**

Yes.

**Strengths And Weaknesses:**

## Strengths

The discussion of the experimental results was thorough and contained interesting insights. For instance, I especially liked the explanation of why the IBS-Admin metric was insensitive to changes in censoring (IBS-family paragraph, bottom right of pg. 5).

The final Discussion section also gave a very helpful summary of the practical takeaways. I also appreciated the additional positioning discussion, which helped to motivate the work in the context of recent other works which also explore the behaviors and limitations of different survival metrics under censoring. Especially in light of this discussion, the topic explored by the paper seems well-motivated/relevant and the practical insights could be helpful to practitioners, especially if some of the weaknesses below are addressed. In general, the paper was well-written.

## Weaknesses

The set of models tested in the experiments is somewhat limited. Especially given that this is a machine learning venue, I would have liked to see more baselines coming from the ML community, especially deep models. (Non-exhaustive) Examples include:

>Katzman, Jared L., et al. "DeepSurv: personalized treatment recommender system using a Cox proportional hazards deep neural network." BMC medical research methodology 18.1 (2018): 24.

>Nagpal, Chirag, et al. "Deep cox mixtures for survival regression." Machine Learning for Healthcare Conference. PMLR, 2021.

Another cause for concern is the use of a Student t-test to compare models across cross validation folds. There is not full independence between the CV folds (e.g., models trained on two different folds will have a significant overlap in training data), meaning the $\Delta_k$ values are not independent which may invalidate the t-test. This should at least be discussed if not corrected for.

Results for Uno's C-index are provided in the appendix, and empirically this metric has qualitatively similar behavior to the others. However, given that Uno's C-index is specifically designed to handle the problem addressed by the paper (the dependence of Harrell's C-index on the censoring distribution), it should be included in the list of metrics discussed in Section 2.2.

My primary concern is that it is not clear whether the experimental design has fully disentangled the effect of different censoring rates and mechanisms purely on the evaluation metrics themselves. This is because the models that are used for evaluation are also trained with variable censoring mechanisms, and this is tied to (in fact equal to) the censoring mechanism which impacts the metric calculation. To put it another way, one could imagine a 2D space, where different points on the x-axis represent different survival models, and different points on the y-axis represent different censoring mechanisms used to compute the metrics. In this metaphor, the paper is only evaluating scenarios on the line y=x (metric censoring mechanism = censoring mechanism which impacts the model), rather than exploring the full 2D space.

Of course, in practice, this will be the case: we will have to train and evaluate models using data with a single censoring mechanism. However, it is possible that in the future we will have a new class of survival models which are impacted differently by censoring than current models. In this case, the coupled model change + metric change due to censoring may not be represented in the current experiments. It therefore seems valuable to try to more completely disentangle the effect of censoring on the metrics in isolation, without also varying the model.

---

> ### Author Rebuttal · Authors · 2026-03-30
>
> We thank the reviewer for the constructive feedback. The comments are helpful, and we address them below.
>
> **Weakness 1**
> We agree that, for an ML venue, testing additional deep baselines is valuable. Our main panel already includes two neural survival models (DeepHit and NMTLR), together with parametric and non-parametric baselines, because our goal is not to build a leaderboard but to study how evaluation metrics behave under censoring across representative model families.
>
> Following your suggestion, we additionally tested DeepSurv and Deep Cox Mixtures in the same pipeline. In our setting, however, they were substantially less stable: training was highly sensitive to the dataset, hyperparameters, and fold, with convergence failures or numerical instability on some datasets (notably PBC and FLCHAIN), especially under nested cross-validation and at high censoring levels. This made them difficult to include in a large-scale comparison. When they did converge (e.g., on METABRIC), the qualitative conclusions were unchanged: we recovered the same broad trends in bias and ranking behavior across metric families, although the magnitude of distortions could vary somewhat, especially for IBS-type metrics. We will clarify this rationale in the revision.
>
> **Weakness 2**
> We fully agree that the standard paired Student t-test is not strictly valid under cross-validation because fold-level estimates are not independent. Following your comment, we replaced the original test with the cross-validation-corrected t-test of Nadeau and Bengio (2003), using fold-wise paired differences and the corrected variance
> $$
> Var_{corr} = \left(\frac{1}{n} + \frac{n_{test}}{n_{train}}\right)s^2
> $$
> This yields a more conservative and CV-appropriate test.
>
> After correction, we recover the same qualitative conclusions. Across the 225 metric × censoring-rate × mechanism configurations, the number of model pairs that are significant in both ST and OR decreases from 26,650 (original) to 13,980 (52.5\%) as expected, but the global-ranking conclusion changes in only 6/225 settings (219/225 preserved). Thus, the correction mainly reduces coverage, not the conclusions themselves. We will update the paper accordingly and note that significance testing under CV remains delicate; repeated CV, bootstrap, or Bayesian alternatives may provide more power, but were beyond the scope of this already large-scale study.
>
> **Weakness 3**
> We agree. Uno’s C-index is especially relevant here. We will move it from the appendix into Section 2.2 and discuss it explicitly in the main text.
>
> **Weakness 4**
> Our experiments operate on the matched line of the 2D space you describe: the censoring regime affecting training and the censoring regime used in evaluation are the same. This is a deliberate choice, not a hidden confound, because the ST/OR comparison is performed on the same trained model and the same predictions; only the test outcomes differ (censored observations in ST vs. fully observed event times in OR). Model parameters are fixed. Therefore, for a given trained model, the ST–OR gap isolates the distortion induced by censoring in the metric computation, not a retraining effect.
>
> We agree, however, that our paper does not cover the full off-diagonal space where training and evaluation censoring differ. Such settings are possible, but are relatively uncommon in practice: survival studies are almost always conducted under a single censoring mechanism, making the matched setting the most realistic one. Variations within a mechanism may occur, but are also less typical. We will state this limitation explicitly. Still, we believe the current design fully supports the main claim of the paper: in the realistic matched setting, censoring can substantially distort reported metric values and downstream ranking conclusions.
>
> **Key question**
> Our intuition is the following. At low-to-moderate censoring, the standard IBS can be slightly pessimistic relative to OR because censoring removes part of the event information without yet fully collapsing the effective integration window. At high censoring, the behavior reverses: the effective estimable horizon contracts, and IBS becomes increasingly dominated by earlier times, where prediction errors are usually smaller. This yields artificially optimistic standard scores relative to OR, hence the sign change. The fact that we observe a similar pattern for both IPCW- and pseudo-observation-based IBS variants suggests that this is not just a specific weighting artifact, but reflects a broader loss of temporal identifiability under heavy censoring. We will make this explanation more explicit in the revision. We stress, however, that this is only an intuition rather than a formal explanation; a deeper understanding of this complex metric would require further dedicated analysis, which we see as a valuable direction for future work and can mention in the discussion.
>
> We thank the reviewer again for the helpful comments.

---

> > ### Author Rebuttal · Reviewer_k2fd · 2026-04-02
> >
> > Thanks to the authors for the detailed response. I believe that Weaknesses 2 and 3 have been fully resolved, and the intuition provided for the Key Question is reasonable and the discussion is appreciated. Using the CV-corrected t-test was especially important, and I am glad to see the candid discussion of what changed (about half as many significant model pairs) but that the overall qualitative results are the same. Based on these improvements, I have increased my score. I will refrain from increasing it further because while I believe that the arguments provided re: Weaknesses 1 and 4 have some merit, I still believe adressing these concerns in full would be necessary for showing that the results of the paper are far-reaching/significant in scope.

---

> > > ### Author Response · Authors · 2026-04-04
> > >
> > > Thank you again for the careful reading and for increasing your score. We are glad that Weaknesses 2 and 3 are now fully resolved.
> > >
> > > Regarding Weakness 1, we have now fully addressed it by adding the two deep models you suggested, DeepSurv and Deep Cox Mixtures, and rerunning the experiments across all datasets. After further stabilizing the training pipeline, in particular by adding stricter numerical safeguards and pruning unstable configurations, we were able to obtain results for both models across all datasets. We have regenerated the two main figures of the paper (Figure 1 and Figure 2) with these additional models included, as well as the table summarizing the hyperparameter search spaces used to optimize these two models, and we provide them here via an anonymous supplementary link that does not reveal reviewer identity: [anonymous supplementary link](https://anonymous.4open.science/r/icml2026-5E32/). As you can directly verify from these updated figures, the overall qualitative trends remain unchanged. What changes slightly is the magnitude of some effects (e.g., variance and sometimes median values), but the main conclusions are preserved. This also holds for the ranking analysis: adding these two models leaves the same overall ranking-preservation conclusions unchanged, which we were glad to confirm. We would be happy to incorporate these results into the paper.
> > >
> > > Regarding Weakness 4, we fully understand your point of view and agree that exploring the full off-diagonal space, where the censoring regime affecting training differs from the censoring regime used for evaluation, would be valuable. However, this would require a new experimental framework and an additional set of experiments, which goes beyond what we could complete within the rebuttal period. We nevertheless still believe that the current matched-setting design is the most practically relevant one, as you also noted: in most real survival applications, training and evaluation are indeed conducted under a single censoring regime. We will therefore keep this point explicit in the paper as a limitation and mention cross-regime evaluation as an important direction for future work.
> > >
> > > We thank you again for the constructive comments, which have clearly strengthened the paper.

---

### Decision · Program_Chairs · 2026-04-30

**Decision:**

Accept (regular)

**Comment:**

The authors empirically study the impact of censoring on different evaluation metrics for survival models.

All the reviewers agree that this paper tackles an important question and is novel. Reviewers at large also appreciated the experimental designs and the careful evaluation.
The main negative comments centered around:
1) The paper conducts a t-test on cross-validated risks without taking into account the fact of the dependence between the folds. This was addressed during the rebuttals in a way that satisfied all the reviewers. (Reviewer k29d and LEEj)

2) Top-1 Ranking Instability: The initial claim regarding Top-1 model selection fragility treated any swap in rank as a failure, irrespective of statistical significance. Additional experiments during the rebuttal showed that, indeed, in many settings where swapping would occur, the top 2 models were not statistically different. The reviewers found that a more careful discussion about this point was warranted. (Reviewer LEEj and YRCs)

3) One reviewer (k2fd) expressed the concern that more experiments could be done.

However, all the reviewers found the rebuttal convincing, and there is a broad consensus that this paper is a solid contribution.